



Atmospheric
Chemistry
and Physics

# Dynamic projection of anthropogenic emissions in China: methodology and 2015–2050 emission pathways under a range of socio-economic, climate policy, and pollution control scenarios

Dan Tong[1], Jing Cheng[1], Yang Liu[1], Sha Yu[2], Liu Yan[1], Chaopeng Hong[1], Yu Qin[3], Hongyan Zhao[1], Yixuan Zheng[1], Guannan Geng[3], Meng Li[1], Fei Liu[3], Yuxuan Zhang[1], Bo Zheng[3], Leon Clarke[4], and Qiang Zhang[1]

[1]Ministry of Education Key Laboratory for Earth System Modelling, Department of Earth System Science, Tsinghua University, Beijing 100084, People's Republic of China
[2]Joint Global Change Research Institute, Pacific Northwest National Laboratory, University Research Court, College Park, MD 20742, USA
[3]State Key Joint Laboratory of Environment Simulation and Pollution Control, School of Environment, Tsinghua University, Beijing 100084, People's Republic of China
[4]Center for Global Sustainability, School of Public Policy, University of Maryland, College Park, MD 20742, USA

**Correspondence:** Qiang Zhang (qiangzhang@tsinghua.edu.cn)

**Abstract.** TS1 Future trends in air pollution and greenhouse gas (GHG) emissions for China are of great concern to the community. A set of global scenarios regarding future socio-economic and climate developments, combining shared socio-economic pathways (SSPs) with climate forcing outcomes as described by the Representative Concentration Pathways (RCPs), was created by the Intergovernmental Panel on Climate Change (IPCC). Chinese researchers have also developed various emission scenarios by considering detailed local environmental and climate policies. However, a comprehensive scenario set connecting SSP–RCP scenarios with local policies and representing dynamic emission changes under local policies is still missing.

In this work, to fill this gap, we developed a dynamic projection model, the Dynamic Projection model for Emissions in China (DPEC), to explore China's future anthropogenic emission pathways. The DPEC is designed to integrate the energy system model, emission inventory model, dynamic projection model, and parameterized scheme of Chinese policies. The model contains two main modules, an energy-model-driven activity rate projection module and a sector-based emission projection module. The activity rate projection module provides the standardized and unified future energy scenarios after reorganizing and refining the outputs from the energy system model. Here we use a new China-focused version of the Global Change Assessment Model (GCAM-China) to project future energy demand and supply in China under different SSP–RCP scenarios at the provincial level. The emission projection module links a bottom-up emission inventory model, the Multi-resolution Emission Inventory for China (MEIC), to GCAM-China and accurately tracks the evolution of future combustion and production technologies and control measures under different environmental policies. We developed technology-based turnover models for several key emitting sectors (e.g. coal-fired power plants, key industries, and on-road transportation sectors), which can simulate the dynamic changes in the unit/vehicle fleet turnover process by tracking the lifespan of each unit/vehicle on an annual basis.

With the integrated modelling framework, we connected five SSP scenarios (SSP1–5), five RCP scenarios (RCP8.5, 7.0, 6.0, 4.5, and 2.6), and three pollution control scenarios (business as usual, BAU; enhanced control policy, ECP; and best health effect, BHE) to produce six combined emission scenarios. With those scenarios, we presented a wide range of China's future emissions to 2050 under different development and policy pathways. We found that, with a combination of strong low-carbon policy and air pollution control

**Published by Copernicus Publications on behalf of the European Geosciences Union.**

policy (i.e. SSP1-26-BHE scenario), emissions of major air pollutants (i.e. $SO_2$, $NO_x$, $PM_{2.5}$, and non-methane volatile organic compounds – NMVOCs CE1) in China will be reduced by 34 %–66 % in 2030 and 58 %–87 % in 2050 compared to 2015. End-of-pipe control measures are more effective for reducing air pollutant emissions before 2030, while low-carbon policy will play a more important role in continuous emission reduction until 2050. In contrast, China's emissions will remain at a high level until 2050 under a reference scenario without active actions (i.e. SSP3-70-BAU). Compared to similar scenarios set from the CMIP6 (Coupled Model Intercomparison Project Phase 6), our estimates of emission ranges are much lower than the estimates from the harmonized CMIP6 emissions dataset in 2020–2030, but their emission ranges become similar in the year 2050.

## 1    Introduction

The rapid development of China has led to severe air pollution due to the ever-increasing energy demand and lax environmental legislation over the past decades, and this exerts negative influences on human health, climate, agriculture, and ecosystems (Liu et al., 2019; Xue et al., 2019a; Zheng et al., 2019). In 2013, China implemented the Air Pollution Prevention and Control Action Plan (denoted the Action Plan) to fight against air pollution (China State Council, 2013), and a series of active clean air policies for various sectors were adopted in support of the Action Plan. Consequently, the emissions of major air pollutants have decreased, and the air quality has substantially improved since 2013 (Zheng et al., 2018; Cheng et al., 2019; Geng et al., 2017, 2019; Xue et al., 2019b; Zhang et al., 2019a). The national annual mean $PM_{2.5}$ concentrations decreased from $72 \, \mu g \, m^{-3}$ in 2013 to $43 \, \mu g \, m^{-3}$ in 2017 (Ministry of Ecology and Environment, MEE, 2014, 2018). However, the $PM_{2.5}$ concentration is still higher than China's air quality standard of $35 \, \mu g \, m^{-3}$ and the World Health Organization (WHO) $PM_{2.5}$ guideline value of $10 \, \mu g \, m^{-3}$ (WHO, 2006). In support of continuous air quality improvement, a series of policies, such as the Three-Year Action Plan for Winning the Blue Sky Defense Battle (the Three-Year Plan) promulgated in 2018 (China State Council, 2018), have been executed. Additionally, China is committed to low-carbon economic development to achieve its nationally determined contribution (NDC) target and contribute to limiting global warming (United Nations Framework Convention on Climate Change, 2015), which could cause substantial reductions in both air pollution and GHG emissions. Thus, future air pollution and GHG emission trends in China are of great concern to the community.

Future changes in energy and emissions in China are either projected separately or incorporated into Asia within global scenarios (Cofala et al., 2007; O'Neill et al., 2010; Amann et al., 2013; Rao et al., 2017; Gidden et al., 2019).

Global scenarios, such as the new generation of global scenarios combining shared socio-economic pathways (SSPs) with climate forcing outcomes as described by the Representative Concentration Pathways (RCPs), can reflect plausible future emissions based on socio-economic, environmental, and technological trends at the regional scale (Rao et al., 2017). However, there are several challenges in using these global scenarios in China's case. First, due to the incomplete knowledge of China's local policies, current global scenarios lack detailed descriptions of national and local energy and pollution control policies. Second, by employing simple extrapolation to emission factors, future estimates from the global scenarios could not provide the complete evolution of future combustion–production technologies and emission control measures. Third, recent emissions in China have changed dramatically as a consequence of clean air actions (Zheng et al., 2018), while historical emission data used in the global scenarios cannot easily capture the fast changes in emissions during recent years or over the next several years in China (Hoesly et al., 2018).

Previous studies have investigated future emission trends in China by considering detailed local policies (Wei et al., 2011; Xing et al., 2011; Zhao et al., 2013; Shi et al., 2016; Jiang et al., 2018; N. Li et al., 2019). These scenarios describe future emission changes based on a set of assumptions that reflect China's economic growth, energy demand, up-to-date air quality, and climate mitigation policies. However, most of these local scenarios are disconnected from global scenarios (e.g. SSP–RCP scenarios; Rao et al., 2017), and only a few scenarios are comparable with IPCC scenario sets (Jiang et al., 2018). Usually, these local scenarios neglect the linkage between Chinese and global development pathways, and air pollutant and $CO_2$ emissions are investigated separately. Moreover, these studies could not simulate the process of new technology entering the markets due to a lack of technology-based projection models. In addition, for other researchers, further studies like air quality and health impact analysis based on these scenarios are difficult to carry out because of the absence of public emission data products.

In this work, with the motivation to build a comprehensive scenario set that connects global scenarios with local policies and represents dynamic emission changes under local policies, we developed a dynamic projection model for China's future anthropogenic emissions, named the Dynamic Projection model for Emissions in China (DPEC). The DPEC is designed to track the dynamic changes in emissions, future combustion–production technologies, and emission control measures. This model includes an energy-model-driven activity rate projection module and a sector-based emission projection module, which integrates the energy system model, emission inventory model, dynamic projection model, and parameterized scheme of Chinese policies. Based on the DPEC, we created six emission scenarios by connecting five SSP scenarios (SSP1–5), five RCP scenarios (RCP8.5, 7.0, 6.0, 4.5, and 2.6), and three pollution control

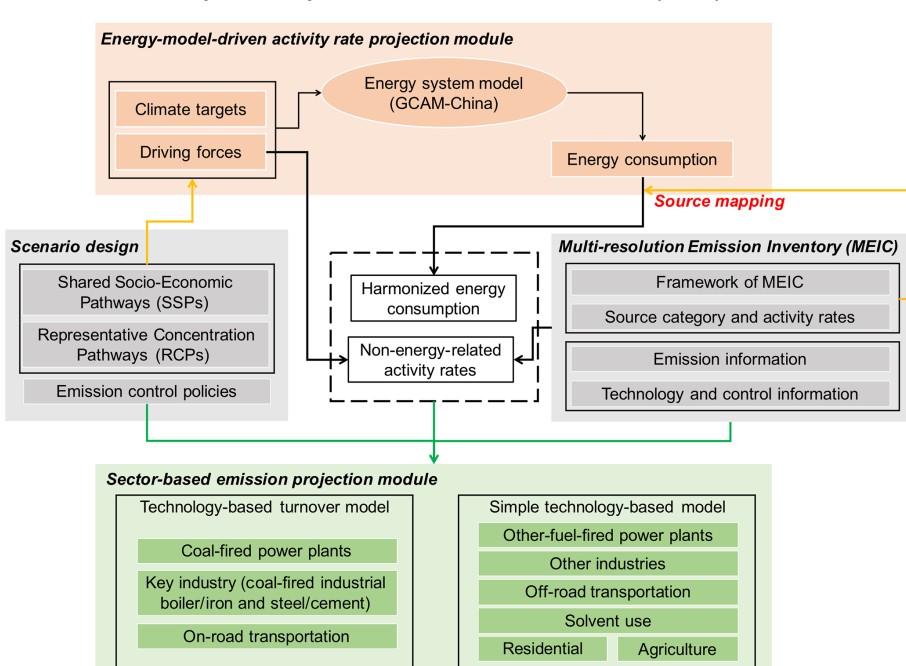

**Figure 1.** Framework of the Dynamic Projection model for Emissions in China (DPEC).

scenarios (business as usual, BAU; enhanced control policy, ECP; and best health effect, BHE) to explore future emission pathways during 2015–2050. Finally, we compared our estimates with similar scenarios from the harmonized Coupled Model Intercomparison Project Phase 6 (CMIP6) emissions dataset (Gidden et al., 2019). In this work, the purposes of developing the DPEC and creating a new set of Chinese scenarios are as follows: (1) connect with the IPCC scenario assembly, (2) synthetically consider region-specific and sector-based local policies, (3) develop technology-based turnover models for key emitting sectors to simulate the dynamic changes in future technologies, and (4) provide a set of emission projection datasets to the community. The development of this dynamic model and associated scenarios aims to identify win-win measures and pathways to support the future's short- and long-term synergizing actions on the environment and climate for policymakers.

## 2 Dynamic Projection model for Emissions in China (DPEC)

### 2.1 Model framework

As shown in Fig. 1, the DPEC includes two main modules, an energy-model-driven activity rate projection module and a sector-based emission projection module. The model integrates the energy system model, bottom-up emission inventory model, dynamic projection model, and parameterized scheme of environmental standards and policies.

The energy-model-driven activity rate projection module is set up to produce standardized and unified future activity rates by linking the energy system model with the emission inventory model. The DPEC is developed starting with the bottom-up framework of the Multi-resolution Emission Inventory for China (MEIC) model (available at http://www.meicmodel.org/, last access: 14 April 2020), which also provides us with historical activity rates, technologies, and emission information for each emitting source. MEIC uses a technology-based methodology to calculate air pollutants and $CO_2$ emissions for more than 700 anthropogenic emitting sources for China from 1990 to present, as described in detail in earlier papers (Zhang et al., 2009; Lei et al., 2011a; Zheng et al., 2014, 2018; Liu et al., 2015; Li et al., 2017; M. Li et al., 2019; Tong et al., 2018a). The China-focused version of the Global Change Assessment Model (GCAM-China; see Sect. 2.2.1.) is adopted to provide future energy demand and supply in China under different socio-economic and energy scenarios at the provincial level. The linkage and harmonization of energy outputs is to fit the MEIC source categories by reorganizing and refining the outputs from GCAM-China (Sect. 2.2.2. and 2.2.3.).

The sector-based emission projection module is built to dynamically track the evolution of future combustion–production CE2 technologies and control measures by parameterizing different environmental regulations and policies. Emission sectors included in this projection model are identical to those in the MEIC model. Here emission sources from the MEIC model are commonly classified into six sectors:

power, industry, residential, transportation, solvent use, and agriculture (Sect. 2.3).

Future emissions from each emission source in each province are estimated as follows:

$$E_{i,j,k} =$$
$$A_{i,j,k} \times \sum_m \left( X_{i,j,m} \times EF_{i,j,k,m} \times \sum_n \left( C_{i,j,m,n} \left( 1 - \eta_{k,n} \right) \right) \right),$$
(1)

where $i$ represents the province, $j$ represents the emission source, $k$ represents the air pollutants or $CO_2$, $m$ represents the technologies for manufacturing, and $n$ represents the technologies for air pollution control. The emission ($E$) is estimated by the product of activity rate ($A$), technology distribution ratio ($X$), unabated emission factor (EF), penetration rate ($C$), and removal efficiency of a specific pollution control technology.

## 2.2 Energy-model-driven activity rate projection module

### 2.2.1 The GCAM-China model

Future energy demand and supply is usually provided by integrated energy system models. Several energy system models, such as GCAM (Clarke et al., 2008, 2018; Collins et al., 2015; http://www.globalchange.umd.edu/gcam/, last access: 25 March 2020), the Integrated Model to Assess the Global Environment (IMAGE; Alcamo et al., 1996; Braspenning Radu et al., 2016; https://models.pbl.nl/image/index.php/Welcome_to_IMAGE_3.0_Documentation, last access: 25 March 2020), and the Model of Energy Supply Systems And the General Environmental Impact (MESSAGE; Miketa et al., 2006; Zhang et al., 2019b; http://pure.iiasa.ac.at/id/eprint/1542/, last access: 25 March 2020), are widely used by researchers and policymakers.

GCAM is a global partial equilibrium model with 32 energy–economy regions representing the behaviour and interactions among five systems: the energy system, water, agriculture and land use, economy, and climate. GCAM is stewarded by the Joint Global Change Research Institute (JGCRI) (GCAM, 2019, http://jgcri.github.io/gcam-doc/index.html, last access: 25 March 2020), and more detailed documentation on GCAM can be found at http://www.globalchange.umd.edu/models/gcam/ (last access: 25 March 2020). Over time, GCAM has been increasingly used in climate (Zhou et al., 2013; Fawcett et al., 2015; Calvin et al., 2019; Sinha et al., 2019), energy (Belete et al., 2019; Silva Herran et al., 2019; Wang et al., 2019), land use (Dong et al., 2018; Turner et al., 2018; Vittorio et al., 2018), and modelling studies. In addition, GCAM also provides a number of scenarios and assessments for various organizations and reports, such as the Energy Modeling Forum (EMF),

the U.S. Climate Change Technology Program, the U.S. Climate Change Science Program (Edmonds and Reilly, 1982, 1983; Edmonds et al., 1984), and the IPCC assessment reports (Reilly et al., 1987; Pachauri et al., 2014; Calvin et al., 2017). Most energy system models, including GCAM, take China as a whole section in simulations and fail to reflect the differences in regional or provincial energy and socioeconomic developments. To better explore the provincial evolution and development in China, JGCRI developed and expanded GCAM to include greater spatial detail in China's provinces, and this model is referred to as GCAM-China (Yu et al., 2019). In GCAM-China, the 31 provinces are operated as explicit regions within the global GCAM model. Energy transformation and end-use demand processes are simulated at provincial levels. Therefore, we used GCAM-China (version 4.3) to project future energy demand and supply in China under different economic-energy scenarios at the provincial level.

### 2.2.2 Linkage between GCAM-China and MEIC emission model

To better illustrate the emission characteristics, emission models, such as the MEIC model, always have much more elaborate emission sources and fuel type categories than energy models. Here, we downscaled and disaggregated the GCAM-China outputs and matched them to the MEIC framework (Table S1). The 227 fundamental emission categories in the MEIC model are composed of intercombinations of seven major sectors (including power, heating, industry, residential, transportation, solvent use, and agriculture) and various fuels and productions (Table S1). Different technologies would further divide these emissions into 745 detailed sources; however, the evolution of technology distributions is simulated in the DPEC. Therefore, the interaction data system mainly conducts sector mapping and fuel mapping to match the GCAM-China outputs to the 227 MEIC categories. Except for the heating sector, all the linkage works were operated at the provincial level.

From the sector perspective, the energy and economic-related outputs from the GCAM-China or GCAM model can be divided into four parts: resource production (primary energy), energy transformation (electricity, heat, refining, and other energy transformation), final energy use (buildings, industry, and transportation), and socio-economics (population and gross domestic product, GDP). The power and heating sector in DPEC were linked from the energy transformation part in GCAM-China. The power sectors of the two models have basically no gaps and can be directly matched. Due to the unavailability of provincial energy information in the heating sector in GCAM-China, we first matched the national heat outputs from the GCAM-China energy consumption sector to DPEC and then downscaled to the provincial level with district heat outputs. When downscaling, the heating industrial (to offer thermal energy) in the DPEC is

derived from industry district heat in the industry sector, and residential heating (refers to centralized heating) in the DPEC is obtained from the commercial and residential urban district heat in the building sector. The residential, industry combustion, and transportation sectors in DPEC are matched with the final energy-use parts in GCAM-China, which are building, industry, and transportation. Similar to the power sector, the cement industry and transportation sources could also be seamlessly connected in the two models. Residential in DPEC is taken from the building sector, including cooking and heating. Boilers and kilns are two major industrial combustion sources in DPEC. Activity rates of industrial boilers and cement kilns in DPEC are provided by industry final energy use and cement energy consumption in GCAM-China, respectively. For other industrial kilns (brick and lime), the activity rates are estimated by their base-year data and the future cement energy-use curve. Future iron and steel manufacturing would be simultaneously affected by energy transformation and socio-economic development. Due to the lack of iron and steel projections in GCAM-China (version 4.3), we estimated future iron and steel productions with projected GDP using the elastic coefficient method (Cao et al., 2016) and fixed furnace fractions (electric, coal-fired, gas-fired, and other-fuel-fired) of newly built capacities to maintain the same energy structure as that of the whole industry sector. The activity rates of non-energy-related sectors in the DPEC, including industrial processes CE3, solvent use, and agriculture, were mostly driven by socio-economic outputs from GCAM-China, which were specifically described in Sect. 2.3.

In terms of fuel mapping, the fuel types in GCAM-China include coal, liquids, gas, biomass, solar resources, nuclear, wind resources, geothermal, and hydro-energy, while the MEIC model partitions each fossil fuel or biofuel in a more detailed manner (Table S1). A fuel type ratio database is established based on each specific fuel structure of coal, liquids, gas, and biomass in the MEIC model, and the energy outputs of GCAM-China are then distributed by this base-year proportion to show the detailed fuel use in DPEC. The standard coal equivalent is adopted with the different units of different fuel types. This reaggregation process of fuel types indicates that the absolute amount and relative proportion of coal, liquids, gas and biomass will evolve under the driver of the energy model, while the substructure inside each major fuel type will remain the same as the 2015 levels in MEIC.

### 2.2.3 Harmonization of energy consumption for the year 2015

Eliminating discrepancies in the base year between MEIC and GCAM-China models is pivotal to project future emissions, which can maintain the consistency with energy outputs to the best extent. In this study, 2015 is chosen as the base year, the historical energy and activity rates in 2015 used in MEIC are obtained from China Energy Statistical Year-book (National Bureau of Statistics (NBS), 2016), and the 2015 information in the GCAM-China model is projected, as GCAM is calibrated in 2010 using the historical datasets from the International Energy Agency (IEA, 2011). The deviation ranges of major fuel types in the base year vary from 6 % to 13 %, and these discrepancies are mainly caused by different statistical methods and raw data sources (Hong et al., 2017). These balances should also evolve with the projected future trends instead of remaining unchanged. Thus, we harmonized the GCAM-China energy outputs (which have first been reorganized and downscaled to DPEC fuel type categories) by multiplying the base-year balance ratio (MEIC base-year energy values divided by GCAM-China base-year energy values), rather than add or subtract these balances.

### 2.3 Sector-based emission projection module

Emission projection module is developed for various sectors based on the historical combustion and production technology and emission control information obtained from the MEIC. Given both emission contributions and data availability, the technology-based turnover models built for several key emitting sources (e.g. coal-fired power plants, key industries, and on-road transportation) are used to simulate the dynamic changes in the unit/vehicle fleet turnover process by tracking the lifespan of each unit/vehicle on an annual basis. In addition, technology-based models developed for the remaining emission sources (i.e. other-fuel-fired power plants, other industries, off-road transportation, solvent use, residential, and agriculture sectors) are to directly forecast the effects of different technologies and control measures (Table S2). To develop sector-based projection models with different emission characteristics, we grouped the power and heating sectors into energy supply and divided the industry sector into the industrial combustion and industrial non-combustion sectors.

### 2.3.1 Energy supply

#### Coal-fired power plants

Considering the coal-dominated structure in the power sector and a unit-based power plant database during 1990–2015 developed in the MEIC (Liu et al., 2015; Tong et al., 2018a, b), a unit-based emission projection model for coal-fired power plants was developed in our previous study to assess the evolution of the coal-fired power unit fleet and associated emissions (Tong et al., 2018a). This model was designed to simulate power plant fleet turnover by tracking the lifespan of each power generation unit, which can be used in various future policy analyses and emission estimates for coal-fired power plants. In this work, the total coal power generation and coal consumption are directly obtained from GCAM-China (Table S1). We integrated this already-built projection

model into the DPEC to project future emissions from coal-fired power plants over China through the year 2050.

### Other-fuel-fired power plants

Other fuel types combusted in China's power plants mainly include natural gas and biomass, and their future energy consumption is obtained from GCAM-China (Table S1). A technology-based emission projection model is developed for other-fuel-fired power plants due to limited historical emission information. The emission factors are estimated by projecting the effects of different combustion technologies and end-of-pipe control technologies in the target years (i.e. 2020, 2030, and 2050) according to the environmental policies (Xing et al., 2011; Tian et al., 2013), and the effects in the other years of the future are obtained through linear interpolation.

### Heat plants

District heating is usually supplied by conventional heat plants or combined heat and power (CHP) plants (Rezaie et al., 2012) for industrial and residential purposes. CHP systems are more thermally efficient than producing process heat alone (Lasseter et al., 2004). Therefore, the government promotes the use of CHP plants. In this work, energy consumption in heat plants by fuel type is obtained from GCAM-China (Table S1), and we developed a power technology-based model to project the technology evolution of heat plants. First, we split energy consumption for CHP plants and conventional heat plants according their thermal efficiencies and heat supply policies. Then, we assumed that CHP plants share the same combustion technology and control technology distributions as power plants under corresponding emission scenarios. For conventional heat plants, we simply adopted a similar model that was developed for other-fuel-fired power plants.

### 2.3.2   Industrial combustion

#### Coal-fired industrial boilers and kilns

The coal used in the industry sector is commonly combusted in coal-fired boilers or kilns (i.e. cement, lime, and brick kilns). Coal consumption in boilers is estimated as total industrial coal consumption from GCAM-China minus estimated coal consumption in kilns (Table S1). Future cement coal consumption is obtained directly from GCAM-China. By assuming a simultaneous demand change among these industries of non-metal building materials and similar improvement in energy efficiencies (i.e. energy consumed per unit product), the projections of coal consumption in lime and brick kilns are therefore based on the future trends in cement coal use (Table S1). Thus, the coal consumed in industrial boilers is derived.

A technology-based turnover emission projection model for coal-fired industrial boilers is developed (Fig. S1). The historical information of coal-fired industrial boilers is obtained from the MEE (unpublished data, hereafter referred to as the MEE-boiler database), which includes unit-level boiler capacity, combustion technology, and end-of-pipe control devices. Given that the MEE-boiler database is incomplete, instead of developing a boiler-based emission projection model, we aggregated all industrial boilers into 16 categories based on boiler size ($< 7$, 7–14, 14–24.5, and $\geq 24.5 \, t \, h^{-1}$) and combustion technology (pulverized boiler, circulating fluidized bed, auto-grate, and hand-feed grate) to generate size and technology distributions, which were then used to simulate the turnover of the industrial boiler fleet. As shown in Fig. S1, the model is designed to simulate the operating industrial boiler turnover driven by the demand and retirement policy, assuming that small boilers and outdated combustion technologies are retired early. In a given year, the model first estimates the supply capability of in-fleet boilers after retirement and then estimates the supply gap under the total coal consumption and fills the gap using new coal-fired industrial boilers. Then, we modelled the changes in emission factors by boiler size and combustion technology by considering the evolution of end-of-pipe control technologies under different emission control policies. Note that we separately considered the newly built and old industrial boilers because there are usually different requirements or emission limits for newly built and old industrial boilers.

Emissions from cement, lime, and brick kilns and other associated industrial processes are estimated within the same projection model and are detailed in the industrial non-combustion sector (Sect. 2.3.3).

### Other fuel combustion

The energy consumption by other fuel combustion is directly obtained from GCAM-China (Table S1). The changes in emission factors are estimated by projecting the effects of different combustion technologies and end-of-pipe control measures in the target years (i.e. 2020, 2030, and 2050) according to the environmental policies (Xing et al., 2011), and then these changes are linearly interpolated to the other years in the future.

### 2.3.3   Industrial non-combustion

#### Coke, iron, and steel plants

As the world's largest steel production area, China is reported to contribute almost half of global raw steel production (USGS, 2016). The iron and steel industry involves a series of closely linked processing steps, including preparation of raw materials, iron-making, steel-making, and finishing processes (Wang et al., 2016). First, the productions of sinter, iron, and steel were driven by GDP with a resilience

factor law, similar to the cement projections in GCAM-China (Table S3). For the coke industry, most of the coke is used in the iron-making process; therefore, we projected the coke production based on the change trend in iron production (Table S3). The climate and energy policies would change the capacity structures and electric furnace proportions of these subsectors.

Here, a technology-based turnover model for the iron and steel industry is built. Emissions from each process (sinter, iron, and steel) are independently projected in the model (Fig. S1). The historical unit-based information of each process is also obtained from MEE (unpublished data, hereafter referred to as the MEE-steel database), which includes the unit-based and process-based operational status (when the unit was commissioned and decommissioned), capacity, production, technology type, control devices, and corresponding removal efficiencies. We first aggregated the facilities of each process (sinter, iron, and steel productions) into 32 categories based on years under operation (< 20, 20–40, 40–60, and ≥ 60 years), capacity size (< 0.2, 0.2–1, 1–3, > 3 million tonnes) and current technology (electric furnace and nonelectric furnace). Similar to the model for coal-fired industrial boilers, this model starts by simulating the turnover of steelwork plant fleets. In a given year, the model first estimates the production capability of in-fleet facilities after implementing retirement policies by assuming the early retirement of small and old facilities with outdated technologies and then filling the gap with newly built facilities under the total predicted activity rates. Through fixing the furnace fractions (electric, coal-fired, gas-fired, and other-fuel-fired) of newly built capacities, the ultimately fused capacities shared the same energy structure with the whole industry sector of a given year. Finally, we model the changes in emission factors by considering the evolution of end-of-pipe control technologies for existing and newly built facilities under different environmental regulations. For coke plants, we assumed that all coke is produced in machinery ovens (Huo et al., 2012), and we simply estimated the effects of advanced control measures according to assumed emission standards due to the unavailability of plant-level data.

### Cement plants

China is the world's largest cement producer and consumer (Lei et al., 2011b). To project future emissions from the cement industry, a kiln-based turnover model is built, which is similar to the model built for coal-fired power plants (Tong et al., 2018a). We began with a kiln-based emission inventory for the 1990–2015 period, which provides historical clinker kiln-level technology and emission information including capacity, operating year, production technology, annual production of clinker and cement, and end-of-pipe control devices and corresponding removal efficiencies (Lei et al., 2011b).

A schematic of the model for the cement industry is shown in Fig. S2. The total cement demand is obtained directly from GCAM-China (Table S1). Beginning with the estimated clinker capacity demand, we simulate the year-to-year dynamics of clinker production structure turnover by considering the retirement of outdated kilns (e.g. small, old, or inefficient capacity) and construction of new kilns. The retirement rate of old kilns is driven by the future demand and forced elimination policy of outdated production capacity. A function for ordering the retirement of individual kilns is developed at the provincial level by considering the production technology, age, and designed capacity with descending priority, which is similar to the function created for coal-fired power plants (Tong et al., 2018a). After considering the retired kilns, for a given year, the model then estimates the capacity gap after evaluating the clinker production capacity of in-fleet kilns and fills the gap using newly built kilns. We finally model the changes in kiln-based emission factors by considering the evolution of the end-of-pipe control technologies. To determine the order of end-of-pipe technology upgrades for each individual kiln, we set up an evolution function of end-of-pipe technology at the provincial level by considering the production technology, designed capacity, and age of each kiln (Tong et al., 2018a).

### Other metals and non-metals

Except for coke, iron, and steel plants, as well as cement plants, the emission sources of all other metal products, nonferrous metals, non-metal building materials, and other industrial products from the MEIC are grouped into "other metals and non-metals". Specifically, other metal products and nonferrous metals include foundry products, aluminium, copper, zinc, alumina, and other nonferrous metals. Nonmetal building materials include glass (flat glass and glass products), lime, and brick. Other industrial products mainly include products from the food and drink industry (i.e. bread, cake, biscuits, sugar, beer, wine, and spirits) and the textile industry (i.e. wool, silk, cloth, and synthetic fibres).

The future productions of the above-mentioned industrial products are projected in different ways due to unavailability from the GCAM-China model. The productions of other metal products and nonferrous metals are projected by building the regression models, which are used to describe the relationships among steel production (or GDP) and production of each product based on relevant statistical data from 1990 to 2015 (Table S3). For non-metal building materials, the future productions of flat glass and glass products are estimated based on the annual growth rate of the new building area (Table S3). Lime and brick productions are forecasted by applying the future trends in cement production (Table S1), which are consistent with their coal use projections. To project the productions of the food and drink industry and textile industry products, we built a series of regression models linking per capita GDP with their historical productions (Table S3).

Then, we estimated the changes in emission factors for each production process. In addition to production processes in the food and drink industry and textile industry, we assumed outdated production technologies in other metal industry and non-metal building material industry have similar retirement rates as those from the iron and steel industry and cement industry, respectively. The evolution of different control measures is estimated according to emission standards. For products from the food and drink industry and textile industry, only volatile organic compound (VOC) emissions are considered to be emitted. Because there is no specific production technology provided by the MEIC, we only considered the future evolution of VOC control measures. Here, we simply projected the effects of advanced devices designed to reduce VOCs in the targeted years (i.e. 2020, 2030, and 2050) according to the assumed environmental regulations.

### Petrochemical industry

The petrochemical industry is considered to be the key VOC-related industry, and VOC emissions from 34 types of petrochemical products are estimated in the MEIC inventory. Here, we grouped these products into five subsectors: oil and gas production, distribution, and refining; fertilizer production; solvent production; synthetic materials; and other chemical products.

Specifically, oil and gas production, distribution, and refining include crude oil production, crude oil handling, oil refining, natural gas production, natural gas distribution, oil depots (gasoline and diesel), and oil stations (gasoline and diesel). The activity rates of these industrial processes are projected based on the change rates of energy demands for corresponding fuel types obtained from GCAM-China (Table S1). Fertilizer production includes the production of urea, ammonium bicarbonate, other nitrate fertilizers (e.g. sodium nitrate and calcium nitrate), and NPK (e.g. nitrogen, phosphorus, and potassium) fertilizer. Fertilizers are commonly used in the agriculture sector, and production is determined by fertilizer demand of national crop yield. Therefore, we assumed that the change in fertilizer production is consistent with fertilizer consumption, which is estimated in the agriculture sector (Sect. 2.3.7). Similarly, solvent production is also determined by the market demand, which includes varnish paint, architectural paint, printing ink, and glue production. We projected the production based on the change rates in corresponding solvent use (Sect. 2.3.6).

Synthetic materials mainly include polyvinyl chloride (PVC) products, polystyrene, ethylene, low-density polyethylene (LDPE), high-density polyethylene (HDPE), styrene, polystyrene, vinyl chloride, PVC, propylene, and polypropylene. Each synthetic material is projected by developing the regression models, and they are used to describe the relationship between national GDP and national production based on historical statistical data (Table S3). Other chemical products, including carbon black, sulfuric acid, synthetic

ammonia by coal, pulp, asphalt production, rubber, and tires, are projected using either the change trends in other related products or the regression models linked to GDP (Zhang et al., 2018; Table S3).

For the above-mentioned products, we assumed that the emission factors are only affected by the end-of-pipe control measures due to no specific production technology provided by the MEIC, which is driven by the related environmental policies and emission standards.

### 2.3.4   Residential sector

Total energy consumed by fuel type in rural and urban areas in the residential sector is separately provided by the GCAM-China building sector (Table S1). The residential sector includes two distinct types of coal combustion equipment for different uses (boilers for heating and stoves for cooking and heating), and their emission factors are quite different (Zhang et al., 2007; Peng et al., 2019). The final residential energy-use split for each usage (i.e. residential heating, cooking, and hot water) is provided directly by GCAM-China (Table S1), which is driven by population, building area, and energy service intensity. The split ratio of two combustion technologies (boiler or stove) is exogenous and evolves with specific clean air policies.

A technology-based projection model for the residential sector is developed (Fig. S3). We projected the year-to-year dynamics of coal combustion technologies (boiler or stove) by assuming that coal stoves are used in individual houses for decentralized heating, cooking, and hot water supply, while coal-fired boilers are used for heating in large buildings in urban areas (Zhang et al., 2007). Finally, we projected the effects of clean coal use, advanced coal stoves and boilers, and end-of-pipe control technologies for coal-fired boilers in the target years (i.e. 2020, 2030, and 2050) under different environmental regulation assumptions and then estimated the effects in the other years of the future through linear interpolation.

For fuel types other than coal, due to limited historical information obtained from MEIC, the effects of advanced combustion technologies and control measures are estimated according to their promotion rates based on the assumed environmental policies.

### 2.3.5   Transportation sector

### On-road transportation

There are nine vehicle types contained in the MEIC, including four types of passenger vehicles (heavy-duty buses, HDBs; medium-duty buses, MDBs; light-duty buses, LDBs; and minibuses, MBs) and four types of trucks (heavy-duty trucks, HDTs; medium-duty trucks, MDTs; light-duty trucks, LDTs; and mini-trucks, MTs) as well as motorcycles (MCs). Additionally, passenger vehicles and trucks are further sub-

divided based on four fuel types, including gasoline, diesel, natural gas, and electricity. The provincial-level on-road transportation energy consumption by fuel type is obtained directly from GCAM-China (Table S1).

A vehicle fleet turnover model is developed at the provincial level to simulate future energy consumption and emissions for each vehicle type by tracking the lifespan of each vehicle. As shown in Fig. S4, the model is built to include the vehicle fleet turnover simulation and the evolution of emission factors. For a given year, the model first estimates newly registered vehicles using a back-calculation method based on total on-road energy consumption and historical vehicle registration data (Zheng et al., 2015). Then, the model estimates the number of vehicles that survive (called "in-fleet vehicles") using historical and estimated vehicle registration data. Therefore, we derived the future's vehicle fleet and corresponding energy consumption for each vehicle type. Finally, we modelled the changes in emission factors of in-fleet vehicles, which are estimated by the product of the base emission factors and deterioration correction factor (Zheng et al., 2014). Unabated emission factors of in-fleet vehicles are determined by their registration year and the implementation year of vehicle emission standards (Fig. S5). To avoid double counting in the emission estimation, our model estimated tank-to-well emissions, which means that vehicles using electricity have zero emissions in on-road transportation (Huo et al., 2015).

**Off-road transportation**

Off-road transportation includes agriculture machinery, construction machinery, low-speed trucks, three-wheelers, locomotives, and inland waterways in the MEIC inventory. Because the total energy consumption for off-road transportation in GCAM-China is blended in the industrial sector, we estimated these values exogenously and subtracted them from the GCAM-China industry energy outputs. We assumed that the proportions of on-road and off-road total energy consumptions in the future are the same as the average historical rates during 2010–2015 (the historical rates varied from 0.192 to 0.203, and we used an average rate of 0.198). On the other hand, the electrification ratio of off-road energy was assumed to be similar to that of the on-road sector. Finally, the total energy consumption and structure for off-road transportation are estimated using the historical on-road and off-road split rate, the projected on-road energy consumption, and structures.

At the provincial level, future energy consumption of each off-road type is first estimated based on the annual average growth rate during 2010–2015. Under the constraint of total off-road energy consumption, in a given year, we distributed total energy consumption to each off-road type according their energy consumption shares (Table S1). The changes in emission factors are evaluated according to the reduction proportions caused by the upgrade in emission standards

(Fig. S5). Here, we assumed that the proportion reduction in emission factors between adjacent emission standards in the future is the same as the mean proportion reduction estimated with the published emission standards.

### 2.3.6 Solvent use

Solvent use is identified as one of the major VOC emission sources, which refers to the applications of products containing solvents. Solvents include paints, adhesives, inks, textile coating, pesticides, industrial and domestic cleaning agents, and so on (Wei et al., 2014). In this work, 17 emission sources from solvent use in the MEIC model are further classified into paint use, printing use, pharmaceutical production, vehicle treatment, wood production, pesticide use, and household solvent use.

Paint use includes the paint applied to architecture, vehicles, wood, and other industrial infrastructure (Li et al., 2014, M. Li et al., 2019). The activity rates of different types of paint use are projected by building various regression models (Klimont et al., 2002; Wei et al., 2011). Specifically, architecture interior wall coating and other architecture paint use, as well as paint use for decorative wood and wood furniture, are projected based on the annual growth rate of newly built areas (Table S3). The amount of new car varnish paint and vehicle refurnish paint use is forecasted by developing the regression model linked with the newly registered vehicles and total vehicles, respectively (Table S3). The activity rates of other industry paint use are projected according to the annual growth rate of the above paint use (Table S3).

For solvent use other than paint, printing use (including printing ink and printing cleaning-gasoline solvent) and solvent use for wood production and pharmaceutical production are also forecasted by developing the regression models, and they are used to describe the relationship between national GDP and national amounts based on relevant statistical data from 1990 to 2015 (Table S3). Similar to vehicle paint use, the regression model of passenger vehicle treatment (for de-wax or reseal) is also linked with newly registered vehicles (Table S3). Household solvent use here includes domestic solvent, dry clean $C_2Cl_4$ usage, and glue use, which we estimate through linkage with per capita GDP (Table S3).

Then, the changes in emission factors for various types of solvent use are estimated through the substitution rates of environmentally friendly products (including low-VOC and zero-VOC products) and the effective rates of recovery technologies (e.g. carbon adsorption, incineration, and membrane vapour separations) (Belaissaoui et al., 2016) according to the assumed emission standards and environmental policies.

### 2.3.7 Agriculture

The agricultural sector is distinguished as the main emission source for $NH_3$ emissions (> 90 % of total $NH_3$ emissions),

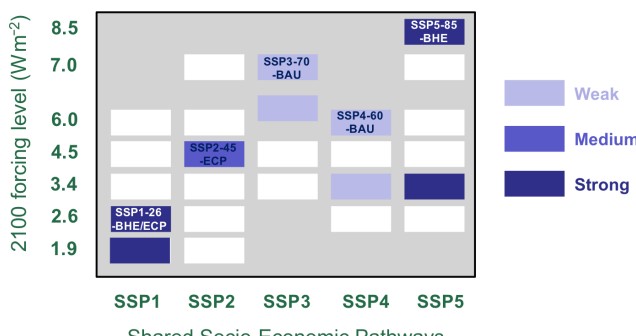

**Figure 2.** The designed scenario ensembles. Three-dimensional constraints, the socio-economic assumptions from the SSPs (SSP1, SSP2, SSP3, SSP4, and SSP5), the climate targets of the RCPs (RCP2.6, RCP4.5, RCP6.0, RCP7.0, and RCP8.5), and the air pollution control ambitions (strong, medium, and weak) from the harmonized CMIP6 emissions dataset were integrated to constitute six China's localized CMIP6 emission scenarios. Each cell in the matrix indicates the feasible scenarios. The nine coloured cells represent the nine scenarios used in the ScenarioMIP experiment ensemble, and labelled cells represent the scenarios we created in this work. This figure is adapted and revised from O'Neill et al. (2016).

including fertilizer use and livestock (Kang et al., 2016). The livestock category includes dairy cattle, other cattle, horses, donkeys, mules, pigs, goats, sheep, and broiler chickens in the MEIC emission inventory. For each type of livestock, we projected the future number of livestock by building the regression model, which is used to describe the relationship between national population and national annual amount based on relevant statistical data from 1990 to 2015 (Table S3). The projected population is obtained from GCAM-China. Then, changes in emission factors are evaluated by assessing the proportions of intensive farming systems (Xu et al., 2017).

Similar to the classification of fertilizer production, the regression model is developed to forecast the total fertilizer application, which is used to describe the relationship between national crop yield and total consumption of fertilizer (Table S3). The future national crop yield is estimated using the product of per capita crop yield and population (Ray et al., 2013). Then, four types of fertilizer use are estimated through multiplying the total fertilizer use by their shares in 2015 (Table S3). The changes in emission factors are modelled by evaluating the different promotion levels of slow-release fertilizer application.

## 3  Scenario design

### 3.1  Definition of scenarios

In this work, five SSP scenarios (SSP1–5; O'Neill et al., 2014) and five RCP scenarios (RCP8.5, 7.0, 6.0, 4.5, and 2.6) are first connected to produce five economic-energy scenarios, namely, SSP1-26, SSP2-45, SSP3-70, SSP4-60, and

SSP5-85. The SSPs were developed over the last several years to describe global developments leading to different challenges for mitigation and adaptation to climate change (O'Neill et al., 2014). And the RCPs were defined by their total radiative forcing (cumulative measure of human GHG emissions from all sources expressed in watts per square metre, $W m^{-2}$) pathway and level by 2100 (van Vuuren et al., 2011). Each SSP–RCP combination represents an integrated scenario of future climate and societal change, which can be used to investigate the mitigation effort required to achieve that particular climate outcome, the possibilities for adaptation under that climate outcome and assumed societal conditions, and the remaining impacts on society or ecosystems (O'Neill et al., 2016; Gidden et al., 2019). Feasible SSP–RCP combinations (cells in Fig. 2) are first identified in terms of meeting mitigation targets for a complete overview of the SSP baseline and climate mitigation scenarios (Rao et al., 2017; Riahi et al., 2017). Among feasible SSP–RCP scenarios, nine scenarios are particularly selected for inclusion in the Scenario Model Intercomparison Project (ScenarioMIP) for CMIP6 (coloured cells in Fig. 2; O'Neill et al., 2016; Gidden et al., 2019). The ScenarioMIP chooses an SSP for each global average forcing pathway based on one or, when compatible, more of the following goals: facilitate climate research, minimize differences in climate, and ensure consistency with scenarios that are most relevant to the IAM (integrated assessment model) and IAV (impacts, adaptation, and vulnerability) communities (O'Neill et al., 2016). In this work, we further selected one scenario from each SSP in the ScenarioMIP experiment ensemble (i.e. SSP1-26, SSP2-45, SSP3-70, SSP4-60, and SSP5-85).

Then, we designed three pollution control scenarios toward medium-term and long-term environmental goals proposed by the government. The first scenario (BAU) is designed to explore the continuous effects of the Action Plan and existing emission standards (before 2015); the second scenario (ECP) is designed to basically attain the grand goal of building a "beautiful China" by 2035 (the State Council of the People's Republic of China, 2019), which requires fundamental improvement in the quality of the environment by achieving its national ambient air quality standards (NAAQS, $35 \mu g m^{-3}$) nationwide until 2030; and the third scenario (BHE) is to ensure a clean environment and maximally protect the public's health, which requires application of best-available technology to eventually attain WHO Interim Target 3 of $15 \mu g m^{-3}$ annual mean $PM_{2.5}$ by 2050.

The BAU scenario assumes that all current environmental legislations and policies released before 2015 would be implemented without any additional environmental policy until 2050. The ECP scenario further considered the emission control policies promulgated, proposed, or likely to be proposed before 2030. Two key control zones are extracted from China to simulate the evolution of pollution controls based on comprehensive consideration of the promulgated policies, geographical locations, and present air pollution conditions. One

zone is the Beijing–Tianjin–Hebei (BTH) and surrounding areas and the Fenwei Plain CE4, and the other is the Yangtze River Delta (YRD). Based on the ECP scenario, the BHE scenario further assumes that the best-available technologies will be phased in and fully applied during 2030–2050 in various sectors (see Sect. 3.3).

Following the interpretation of SSP narratives, a set of assumptions on pollution control is also developed in the CMIP6 database, including a weak pollution control scenario for SSP3 and SSP4, medium one for SSP2, and strong one for SSP1 and SSP5 (Fig. 2, Rao et al., 2017). In our work, the BAU, ECP, and BHE scenarios represent low, central, and high pollution control ambitions, respectively. Following the CMIP6 database framework, we therefore created five air pollution emission scenarios using the five economic-energy scenarios described above, namely SSP1-26-BHE, SSP2-45-ECP, SSP3-70-BAU, SSP4-60-BAU, and SSP5-85-BHE (marked in Fig. 2). Additionally, to explore the benefits of air pollutant emission reductions from mid- and long-term energy transitions, the SSP1-26-ECP scenario is supplemented as the sixth scenario in this work. This combination then represents a range of socio-economic, climate policy, and pollution control scenarios and has been used to investigate the synergistic effects of various future energy developments and emission control policies in this work. Table 1 describes the relevance of the forcing pathway, the rationale for the choice of driving SSP, and the resulting government climate and environmental actions for each emission scenario.

## 3.2 Energy projection

### 3.2.1 Power sector

The total power generation in China has significantly increased by 132.6 % during 2005–2015, which is primarily driven by population growth, industrialization and urbanization (NBS, 2006 and 2016; Tong et al., 2018a). However, up to 71 % of China's power generation was coal-fired in 2015. Given the dominant role of coal-fired power generation, China's government has promoted the development of clean energy in the power sector. Meanwhile, China has also undertaken great efforts to improve the efficiency of coal-fired power units by retiring small and inefficient coal-fired units and building large and high-efficiency units. The optimization of the generation unit fleet mix caused a significant decrease in the coal consumption rate by $91.3 \, \mathrm{gce \, kW \, h^{-1}}$ (where gce is grammes of coal equivalent), which represents 22.4 % of the energy efficiency improvement achieved during 1990–2015 (Liu et al., 2015; Tong et al., 2018a), and the coal consumption rate decreased to $315.4 \, \mathrm{gce \, kW \, h^{-1}}$ CE5.

The energy scenarios adopted in this study reflect different evolution of energy structure and power unit fleet in the power sector. Low radiative forcing targets represent aggressive low-carbon energy transformation required in the future, as well as advanced carbon removal technologies (e.g. car-

bon capture and storage, CCS). Under the SSP1-26 scenario, it is projected that the share of coal-fired electricity will decrease to 14.1 % in 2050 (including 8.3 % of the coal-fired with CCS electricity share) compared to 77.6 % in 2050 under the SSP5-85 scenario. Accordingly, the rapid reduction in coal-fired electricity implies the early retirement of the currently operating coal-fired power capacity in our turnover model. According to estimates, 65 % and 45 % of current coal-fired power capacity needs to be retired early (lifetimes < 40 years) fitting the power structure under the SS1-26 and SSP2-45 scenarios, respectively. The retirement rate for built power units derived from each energy scenario is used to simulate the evolution of the power unit fleet.

Meanwhile, five energy scenarios also reflect various efforts on future energy efficiency improvements. It is projected that the net coal consumption rate decreases of 12.7 %, 12.1 %, 11.3 %, 11.9 %, and 10.4 % during 2015–2050 will be achieved under the SSP1-26, SSP2-45, SSP3-70, SSP4-60, and SSP5-85 scenarios, respectively. The achievement of energy efficiency improvement relies not only on the optimization of the future power unit fleet but also on the application of advanced technologies for newly built units (e.g. ultra-supercritical technology). The penetration rates of advanced technologies are estimated and integrated into our projection model.

### 3.2.2 Industrial sector

During 2005–2015, the energy consumption of China's industrial sector greatly increased by 35.7 % (from 1879.1 to 2922.8 million tce – tonnes of coal equivalent), which is driven by rapid industrial development and ever-increasing demand of energy-intensive products (NBS, 2007 and 2016). In contrast, the energy intensity per unit GDP in the industry sector rapidly decreased during the same period. In recent years, the Chinese government has greatly adjusted the industrial structure by phasing out outdated industrial technologies and capacities, especially in key industries (e.g. coal-fired boilers, steel and iron plants, and cement plants; Zhang et al., 2019a). Meanwhile, China aims to save energy through industrial energy transformation and energy efficiency improvement.

For industrial boilers, China first proposed eliminating coal-fired boilers with capacities smaller than 7 MW by the end of 2017 in the Action Plan. Energy is saved through both the replacement of large high-efficiency coal-fired boilers and switching to other clean-energy-fired boilers. The evolution of boiler fleet turnover under different energy scenarios was fully simulated in our projection model. Similar to the SSP1-26 scenario, the coal used in industrial boilers decreases rapidly, with 54.3 % of coal saved in 2050 compared to 2015. Accordingly, the reduction in coal use would reduce the capacity demand of coal-fired boilers and drive the early retirement (typical lifespan ∼ 20 years) of coal-fired industrial boilers, and it is estimated that the entire coal-fired

**Table 1.** The description of scenarios designed in this study.

| Scenario name | Socio-economic development | Climate policy | Emission control policy | Scenario definition |
|---|---|---|---|---|
| SSP1-26-ECP | SSP1 | RCP2.6 | ECP | Following the heterogeneous and inclusive global developing trends, China would develop sustainably at a reasonably high pace, inequalities are lessened, and technological change is rapid and directed toward environmentally friendly processes, including lower carbon energy sources and high productivity of land, under which societal condition the government is committed to achieving the medium-term environmental targets and modernization goals by 2035. But then, efforts to combat air pollution would basically remain as 2030 level, and there might be more investment in other sustainable development goals, like education and medicine. |
| SSP1-26-BHE | SSP1 | RCP2.6 | BHE | This scenario shares the same SSP forcing pathway with SSP1-26-ECP. Based on achieving the medium-term environmental targets and the modernization goals by 2035, the more optimistic long-term environmental policies and investments are further considered, and the best available technologies are gradually fully applied during 2030–2050, to achieve clean air as in developed countries and maximally protect human health. |
| SSP2-45-ECP | SSP2 | RCP4.5 | ECP | A central pathway in which trends continue their historical patterns without substantial deviations and China's government continues to make strict environmental policies in the short and medium term. |
| SSP3-70-BAU | SSP3 | RCP7.0 | BAU | Inequality and competition among countries would be high and intense. China therefore would develop with a pessimistic trend, and economic growth would slow down while population would increase sharply. To cope with the fierce international competition, little investment would go into education, health, or environment protection, which would lead to the government ignoring the economy and climate issue; meanwhile environmental control would basically stay at the 2015 level. |
| SSP4-60-BAU | SSP4 | RCP6.0 | BAU | Inequality remains high and economies are relatively isolated, with the development in China proceeding slowly. China is highly vulnerable to climate change with limited adaptive capacity due to limited investments, which also lead to limited actions on climate and environmental issues; therefore, environmental control would still remain at the 2015 level. |
| SSP5-85-BHE | SSP5 | RCP8.5 | BHE | The highest future economic increment will be achieved in China under rapid global economic growth. To achieve radical development, climate policies are ignored and highly intensive, fossil-fuel-based energy system would be established, with few advanced technology options. Nevertheless, with rapid economic expansion, the environmental degradation would become more serious, and the government might invest in consistent air pollution controls. Under the accordant global developing trend, the manufacture and end-of-pipe control technologies in China would gradually catch up with developed countries. |

boiler capacity (smaller than 24.5 MW) should be retired by 2050 and 75 % of large coal-fired boiler capacity (larger than 45.5 MW) would be built in our projection model.

Similarly, to reduce energy intensity, the outmoded production technologies would be replaced with more energy-efficient ones in the steel and iron industry and the cement industry. The facility fleet turnover is simulated under various retirement policies created from corresponding energy scenarios. As a result, under the SSP1-26 scenario, to restrict the development of energy-intensive heavy industry, the total production of steel and cement is projected to decrease by 55.3 % and 93.9 % during 2015–2050, respectively.

### 3.2.3 Residential sector

Residential energy consumption in China has steadily increased in the past few decades, driven by increases in total population and building areas (Wang et al., 2014). The total energy consumption in the residential sector increased by 44.9 % from 2005 to 2015, with a 4.2 % annual average growth rate (NBS, 2007, 2016). In recent years, the Chinese government has promoted a series of energy-saving measures to fight against air pollution from the residential sector, including the use of clean energy and clean use of coal (Shen et al., 2019). On the one hand, coal cleaning technologies (e.g. coal washing) are in widespread use, and the coal-washing rate is required to be up to 65 % by the end of 2015 according to the energy development of the 12th FYP. On the other hand, China aimed to switch residential coal to other types of clean energy (e.g. natural gas, electricity, or renewable energy). For instance, by the end of 2017, energy consumption in 6 million households in China (4.8 million households in BTH and surrounding regions) switched from coal to electricity and natural gas (Zhang et al., 2019a).

As estimated in GCAM-China and processed through our energy module, we estimated that approximately 63.0 and 27.3 million households nationwide would switch from coal to electricity and natural gas by 2050 under the SSP1-26 and SSP2-45 scenarios, respectively. Accordingly, 181.3 and 73.9 million tonnes of coal energy are saved by 2050 under the SSP1-26 and SSP2-45 scenarios, respectively, compared to the SSP5-85 scenario. Coal washing can not only substantially reduce air pollution emissions by lowering the ash and sulfur contents in coal but also improve the thermal efficiency. The shares of washed coal during 2015–2050 are estimated according to current energy-saving policies and assumptions of future energy-saving policies under different energy scenarios because GCAM-China cannot reflect these measures due to no specific coal classification. We assumed that a measure of coal washing would continue to be carried out in the future under the SSP1-26 and SSP2-45 scenarios to fit other energy-saving measures under low radiative forcing targets. We estimated that 85 % and 55 % of coal-washing rates would be achieved by 2050 under the SSP1-26 and SSP2-45 scenarios, respectively.

### 3.2.4 Transportation sector

Attributed to the dramatic rise in the total number of vehicles, the energy consumption in China's transportation sector grew 104.9 % in total during 2005–2015 (NBS, 2007 and 2016). Energy consumed in the transportation sector can be saved by improving fuel efficiency and promoting electric vehicles (Wang et al., 2014). China has implemented fuel-efficiency standards for light-duty vehicles since 2004, and an updated standard issued in 2011 requires that the efficiency for passenger cars is up to $14.3\,\text{km}\,\text{L}^{-1}$ by 2015. Meanwhile, China also promotes the development of electric vehicles, and it is reported that the total of electric vehicles reached > 3.1 million by 2019 (The Ministry of Public Security, 2020).

Energy scenarios from GCAM-China are adopted to estimate future vehicle fleet turnover. It is projected that the total vehicle population would be up to 2.64, 2.29, 2.06, 2.34, and 3.25 billion by 2050 under the SSP1-26, SSP2-45, SSP3-70, SSP4-60, and SSP5-85 scenarios, respectively. We found that the share of electric vehicles is as low as 13.8 %, even under the SSP1-26 scenario, which underestimates the future development of electric vehicles in China according China's 13th FYP development planning of electric vehicles. For consistency, we followed the energy structure and related assumptions from GCAM-China in our emission projections. The improvement in fuel efficiencies for each vehicle type reflected in GCAM-China is also estimated though energy consumption and projected vehicle kilometres travelled (VKTs), which has been integrated into our fleet turnover model. As a result, there is a slight but consistent improvement in the average fuel economy. Under the SSP1-26 scenario, fuel efficiency increases from $4.8\,\text{km}\,\text{L}^{-1}$ in 2015 to $5.3\,\text{km}\,\text{L}^{-1}$ in 2050 for heavy-duty diesel vehicles, from $3.8\,\text{km}\,\text{L}^{-1}$ in 2015 to $4.6\,\text{km}\,\text{L}^{-1}$ in 2050 for heavy-duty gasoline vehicles, and from $14.3\,\text{km}\,\text{L}^{-1}$ in 2015 to $25\,\text{km}\,\text{L}^{-1}$ in 2030 for light-duty gasoline vehicles and then remains steady.

### 3.3 End-of-pipe emission control scenarios

### 3.3.1 Power sector

Figure 3 shows the policy evolution under each emission scenario in the power sector. Under the BAU scenario, we assumed that all the coal-fired power plants would follow the emission limits of the standard GB 13223-2011 until 2050. The emission limits for $SO_2$, $NO_x$, and particulates are 100, 100 (200 for existing), and $30\,\text{mg}\,\text{m}^{-3}$, respectively. Under the ECP scenario, China pledged an "ultra-low" emission standard in December 2015, the emission standard is strengthened to 30 for $SO_2$, 50 for $NO_x$, and $10\,\text{mg}\,\text{m}^{-3}$ for particulates, and all the retrofits are achieved nationwide by 2020 and earlier in key regions. Under the BAT scenario, additional recommended BAT values (limits of 20 for $SO_2$, 30 for $NO_x$, and $5\,\text{mg}\,\text{m}^{-3}$ for particulates) from the developed countries are considered after 2030 to achieve the 2050 air

| Emission source | Scenario | Region | 2015 | 2016 | 2017 | 2018 | 2019 | 2020 | 2021 | ... | 2025 | ... | 2028 | 2029 | 2030 | 2031 | ... | 2035 | ... | 2040 | ... | 2045 | 2050 |
|---|---|---|---|---|---|---|---|---|---|---|---|---|---|---|---|---|---|---|---|---|---|---|---|
| Coal-fired power plants | BAU | All regions | GB 13223-2011 [SO₂-100; NOₓ-100/200; PM-30] | | | | | | | | | | | | | | | | | | | | |
| | ECP | BTH & FW Plain | GB 13223-2011 | ultra-low emission standard¹ [SO₂-35; NOₓ-50; PM-10] | | | | | | | | | | | | | | | | | | | |
| | | YRD | GB 13223-2011 | ultra-low emission standard¹ | | | | | | | | | | | | | | | | | | | |
| | | Other regions | GB 13223-2011 | ultra-low emission standard¹ | | | | | | | | | | | | | | | | | | | |
| | BHE | All regions | GB 13223-2011 | ultra-low emission standard¹ | | | | | | | | | | | | | | | BAT recommended value² [SO₂-20; NOₓ-30; PM-5] | | | | |
| Other thermal power plants | BAU | All regions | GB 13223-2011 | | | | | | | | | | | | | | | | | | | | |
| | ECP | BTH & FW Plain | GB 13223-2011 | special limits³ [SO₂-50; NOₓ-100; PM-20] | | | | | | ultra-low emission standard¹ | | | | | | | | | | | | |
| | | YRD | GB 13223-2011 | special limits³ | | | | | | ultra-low emission standard¹ | | | | | | | | | | | | |
| | | Other regions | GB 13223-2011 | special limits³ | | | | | | ultra-low emission standard¹ | | | | | | | | | | | | |
| | BHE | All regions | GB 13223-2011 | special limits³ | | | | | | ultra-low emission standard¹ | | | | | | | BAT recommended value² | | | | |

**Figure 3.** Policy evolution under each emission scenario in the power sector during 2015–2050. The power sector is divided into coal-fired power plants and other thermal power plants. Policies in each emission source are strengthened in the order of blue, green, and orange, and gradient colour reflects the transition from one standard to another during certain years (from a solid line to a dashed line). The superscripted numbers represent different policies or standards, and the same superscripted number represents the same policies or standards applied in various regions.

quality target. The policy evolution of other thermal power plants is similar to that of the coal-fired power plants except the policy effective year. Like under the ECP scenario, the additional special limits were enforced before the ultra-low emission standard between 2018 and 2024.

### 3.3.2 Industrial sector

The industrial sector includes various subsectors, as described above. Figure 4 shows the evolution of emission control policies in different subsectors under three emission scenarios, as we can see, emissions from all the main industries are regulated through national emission standards, except the key VOC-related industries (i.e. the petrochemical industry), by 2015. All the regions would follow these current emission standards until 2050, and there are no specific regulations for key VOC-related industries under the BAU scenarios. Furthermore, under the ECP scenarios, ultra-low emission transformation is assumed to be completed in all industries by the end of 2030 except the key VOC-related industries. We also assumed the key VOC-related industries would reach low emission levels through current mature VOC-removal technologies (Fig. 4). More precisely, based on the ultra-low emission standards, all industries would achieve the BAT recommended values by the end of 2050 under the BHE scenario.

Different emission scenarios reflect to what extent the emission standards are strengthened; here, we comprehensively considered the evolution differentiations among subsectors and regions. First, we assumed that the completion year for each standard or policy in key control zones is a few years earlier than other regions in China according to the promulgated policies. For instance, a policy on ultra-low emission transformation in the iron and steel industry is implemented in 2019, which requires the completion of retrofits using the ultra-low emission technique in key re-

gions (i.e. the BTH and Fenwei Plain, and the YRD region) by the end of 2025. Therefore, we assumed that all the ultra-low emission retrofits would be finished in other regions by the end of 2030. Second, subsectoral differentiation within the same policy is considered. Taking the ultra-low emission standard as an example, we assumed that the ultra-low emission standard would eventually be achieved in all industries. The ultra-low emission standard was first raised in coal-fired power plants, which was implemented in 2016 and would be completed nationwide by the end of 2020 as planned. Hereafter, the ultra-low emission standard for the iron and steel plants was issued in 2019, which required the retrofits to be completed to at least 80 % capacity nationwide by the end of 2025. Following the coal-fired power plants and iron and steel plants, we projected an ultra-low emission standard for cement plants that would then be proposed during 2020–2025, and retrofits would be accomplished by 2030.

### 3.3.3 Residential sector

There is no specific regulation in the residential sector before 2015; therefore, we assumed that emissions from the residential sector are not regulated under the BAU scenario (Fig. S6). Under the ECP scenario, clean coal and advanced stoves have been promoted to reduce emissions in recent environmental policies. We assumed continual upgrades for stoves and coal washing to reach relatively low emission levels through 2030. While under the BHE scenarios, for the long-term air quality target, we supplemented the enhanced controls through innovations of stoves and residential coal stoves until 2050.

### 3.3.4 Transportation sector

Emission reductions from the transportation sector are mainly achieved through fleet turnover in recent years, which means that old vehicles are being replaced by newer, cleaner

| Emission source | Scenario | Region | Policy evolution (2015–2050) |
|---|---|---|---|
| Coal-fired boilers (heating and industrial, residential) | BAU | All regions | GB 13271-2014 [$SO_2$-200/300; $NO_x$-200/300; PM-30/50] |
| | ECP | BTH & FW Plain | GB 13271-2014 → limits[3] → ultra-low emission standard[1] |
| | ECP | YRD | GB 13271-2014 → limits[3] → ultra-low emission standard[1] |
| | ECP | Other regions | GB 13271-2014 → limits[3] → ultra-low emission standard[1] |
| | BHE | All regions | GB 13271-2014 → limits[3] → ultra-low emission standard[1] → BAT recommended value[2] |
| Iron and steel plants (sinter/coke/iron/steel) | BAU | All regions | GB 28662-2012 (sinter) [$SO_2$-200; $NO_x$-300; PM-50] / GB16171-2012 (coke) [$SO_2$-100; $NO_x$-200; PM-50] / GB 28663-2012 (iron) [PM-20; PM-fu-8] / GB 28664-2012 (steel) [PM-20/50; PM-fu-8] |
| | ECP | BTH & FW Plain | standards → special limits[4] [$SO_2$-50/100; $NO_x$-100/320; PM-15/20] → ultra-low emission standard[5] [$SO_2$-35/50; $NO_x$-50/200; PM-10] |
| | ECP | YRD | standards → limits[4] → ultra-low emission standard[5] |
| | ECP | Other regions | standards → limits[4] → ultra-low emission standard[5] |
| | BHE | All regions | standards → limits[4] → ultra-low emission standard[5] → BAT recommended value[6] [$SO_2$-35; $NO_x$-50/100; PM-10] |
| Nonferrous metal | BAU | All regions | GB 31574-201529(Pb,Zn), GB 31575-201530(Cu,Al), GB 25468-201031(Mg), GB 25467-201032(Ni) [$SO_2$-150/200; $NO_x$-200; PM-30/50] |
| | ECP | BTH & FW Plain | standards → special limits[7] [$SO_2$-100; $NO_x$-100; PM-15/20] → ultra-low emission standard[5] |
| | ECP | YRD | standards → special limits[7] → ultra-low emission standard[5] |
| | ECP | Other regions | standards → special limits[7] → ultra-low emission standard[5] |
| | BHE | All regions | standards → special limits[7] → ultra-low emission standard[5] → BAT recommended value[6] |
| Cement plants | BAU | All regions | Cement energy norm; GB 4915-2013 [$SO_2$-200; $NO_x$-400; PM-30] |
| | ECP | BTH & FW Plain | GB 4915-2013 → special limits[8] [$SO_2$-100; $NO_x$-320; PM-20] → ultra-low emission standard[9] [$SO_2$-100; $NO_x$-200; PM-15] |
| | ECP | YRD | GB 4915-2013 → special limits[8] → ultra-low emission standard[9] |
| | ECP | Other regions | GB 4915-2013 → special limits[8] → ultra-low emission standard[9] |
| | BHE | All regions | GB 4915-2013 → special limits[8] → ultra-low emission standard[9] → BAT recommended value[10] [$SO_2$-50; $NO_x$-150; PM-10] |
| Flat glass | BAU | All regions | GB29495-2013 [$SO_2$-400; $NO_x$-700; PM-50] |
| | ECP | BTH & FW Plain | GB29495-2013 → special limits[11] [$SO_2$-200; $NO_x$-300; PM-30] → ultra-low emission standard[5] |
| | ECP | YRD | GB29495-2013 → special limits[11] → ultra-low emission standard[5] |
| | ECP | Other regions | GB29495-2013 → special limits[11] → ultra-low emission standard[5] |
| | BHE | All regions | GB29495-2013 → special limits[11] → ultra-low emission standard[5] → BAT recommended value[10] [$SO_2$-100; $NO_x$-150; PM-10/30] |
| Brick/lime and other industries | BAU | All regions | GB 29620-2013 (brick and lime) [$SO_2$-300; $NO_x$-200; PM-30] |
| | ECP | BTH & FW Plain | standards → special limits[8] → ultra-low emission standard[9] |
| | ECP | YRD | standards → special limits[8] → ultra-low emission standard[9] |
| | ECP | Other regions | standards → special limits[8] → ultra-low emission standard[9] |
| | BHE | All regions | standards → special limits[8] → ultra-low emission standard[9] → BAT recommended value[10] |
| Key VOC-related industries | BAU | All regions | no specific regulations |
| | ECP | BTH & FW Plain | no specific regulations → improve LDAR → further improve LDAR technology; install VOCs control facility in all VOC-related industries → relative low emission levels |
| | ECP | YRD | no specific regulations → improve LDAR → further improve the LDAR technology; install VOC control facility in all VOC-related industries → relative low emission levels |
| | ECP | Other regions | no specific regulations → improve LDAR in chemical industry → further improve the LDAR technology; install VOC control facility in all VOC-related industries → relative low emission levels |
| | BHE | All regions | no specific regulations → improve LDAR → further improve LDAR technology; install VOC control facility in all VOC-related industries → relative low emission levels → Innovation of VOC control facilities |

**Figure 4.** Policy evolution under each emission scenario in the industry sector during 2015–2050. Here the industry sector is divided into seven subsectors (i.e. coal-fired boilers, iron and steel plants, cement plants, nonferrous metal, flat glass, brick–lime, and other industries, as well as key VOC-related industries). Policies in each emission source are strengthened in the order of blue, green, orange, and yellow, and gradient colour reflects the transition from one standard to another during certain years (from a solid line to a dashed line). The superscripted numbers represent different policies or standards, and the same superscripted number represents the same policies or standards applied in various regions.

models subjected to tougher emission standards (Zheng et al., 2018). Therefore, upgrading emission standards plays a vital role in reducing emissions. We modelled the evolution of emission standards for light-duty gasoline vehicles, and heavy-duty gasoline vehicles, light-duty diesel vehicles, heavy-duty diesel vehicles for on-road transportation and off-road transportation (Fig. S5). Under the BAU scenario, we assume that all the registered vehicles comply with the emission standards issued before 2017 and through 2050 with no more stringent emission standards. Therefore, China V emission standards for all on-road vehicles except heavy-duty gasoline vehicles (China IV) are implemented under this scenario. The China III emission standard for off-road transport is implemented. To reduce emissions, further implementation of China VI emission standards for all on-road vehicles and China V for off-road transport is assumed under the ECP scenario. Under the most stringent scenario (BHE scenario), "assumed China VII" emission standards for all on-road vehicles and China VI emission standards for off-road transport would be gradually implemented during 2030–2050.

### 3.3.5 Solvent use

Similar to the VOC-related industries, there are currently no specific regulations for controlling VOC emissions from solvent use. Therefore, we assumed that no effective regula-

tions are implemented under the BAU scenario. Under the ECP scenario, to reach low emission levels of VOCs, we further improved the water-soluble solvent use and installed widespread VOC control facilities in the coating and painting industry. Note that emissions decrease to relatively low levels in the key control zones earlier than in the other regions. Under the BHE scenario, to maximally reduce VOC emissions, we considered the innovations of solvent use and VOC control facilities in the last 5 years before 2050 (2045–2050) according to the best-available technologies from developed countries (European Commission, 2016).

### 3.3.6 Agriculture

Agriculture is one of the least-controlled emission sources in recent years (Zheng et al., 2018). We assumed enhancement of $NH_3$ controls under all emission scenarios except the BAU scenario. Before 2020, we promoted the use of organic fertilizer and resource utilization of poultry excrement and straw. During 2020–2030, we further reduced emissions through the enhancement of intensive cultivation and grazing and the promotion of slow-release fertilizer under the ECP scenario (Pan et al., 2016; Ju et al., 2019). Under the BHE scenario, except for the early achievement of relatively low emission levels nationwide, the innovation of cultivation and grazing is further considered during 2045–2050.

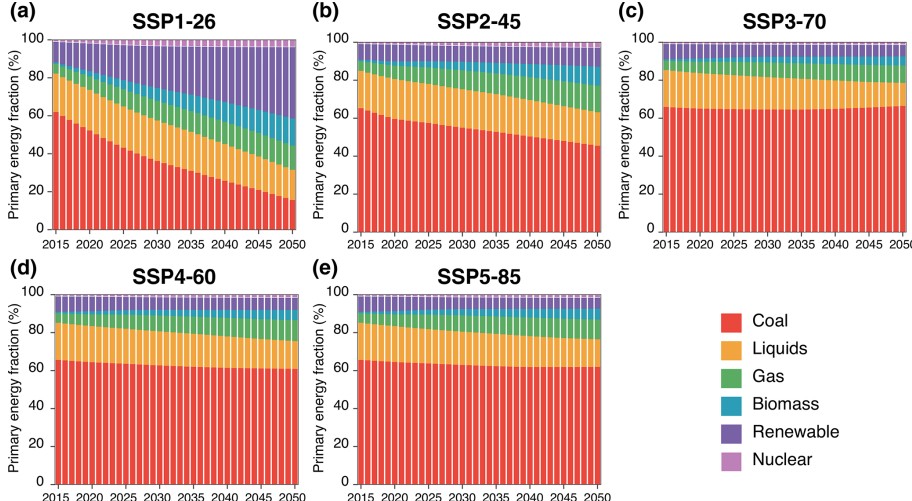

**Figure 5.** Evolution of primary energy structure under different scenarios. The scenarios plotted here include (**a**) SSP1-26, (**b**) SSP2-45, (**c**) SSP3-70, (**d**) SSP4-60, and (**e**) SSP5-85. This figure shows the yearly changes of primary energy (coal, liquids, gas, biomass, renewable, and nuclear) structure under five combined socio-economic–energy scenarios during 2015–2050.

## 4 Results

### 4.1 Evolution of China's future energy system during 2015–2050

Figure 5 shows the yearly evolution of the primary energy structure under five energy scenarios during 2015–2050. At present, coal is the main primary energy source, accounting for more than 60 % of the total primary energy in 2015. Under the lax climate targets, coal will continue to have the dominant role in the future's energy supply structure. We can see a similar future primary energy structure under the SSP3-70, SSP4-60, and SSP5-85 energy scenarios, and the coal fractions are relatively stable until 2050 and close to those in 2015. For the other energy sources, there are obvious increases in the use of gas and biomass sources under the SSP3-70, SSP4-60, and SSP5-85 energy scenarios, in total accounting for 13.9 %, 16.4 %, and 16.1 % in 2050 compared to ∼ 6 % in 2015, respectively. Correspondingly, the fractions of liquids are reduced by 5 %–7 % during 2015–2050. Under the less-stringent climate target (i.e. the SSP2-45 scenario), coal is gradually replaced by gas and biomass, and the coal fraction decreases to 45.5 % by 2050 with an annual reduction rate of 1.0 %. We can see that renewable energy develops very slowly under the SSP2-45 scenario, and only a 1.7 % increase in renewable penetration would be achieved over the next 35 years. Under the stringent climate target of the SSP1-26 energy scenario, the effects include rapid decreases in coal use and increases in renewable energy to limit climate warming. The coal fraction decreases to 15.7 % by 2050, with an annual reduction rate of 4.0 % during 2015–2050. Meanwhile, the renewable fraction increases from 11.0 % in 2015 to 37.7 % in 2050. In China's recent renewable energy development plan, China has proposed the

objective to increase the share of non-fossil-fuel energy in total primary energy consumption to 15 % by 2020 and to 20 % by 2030 (comparable with renewable development projection from the SSP1-26 scenario), which implies determination of energy transformation and low-carbon energy system development for the Chinese government (National Development and Reform Commission, 2016). Additionally, with the climate warming constraints, the CCS technology is gradually applied in the SSP1-26 energy scenario, and coal–CCS accounts for 9.7 % and 16.3 % in all coal-fired applications in the industry and power sectors, respectively, in 2050. However, the CCS is basically not adopted in other energy scenarios. Under all five energy scenarios, nuclear energy contributes very small fractions of total primary energy. This is because the GCAM-China model fixes national nuclear plans by 2030, and limited growth is subsequently considered only in coastal provinces after 2030.

To investigate the changes in sectoral energy consumption, Fig. 6 further shows coal, liquids, and gas consumption in 2020, 2030, and 2050 in the power, industry, residential, and transportation sectors, under five energy scenarios. As shown in Fig. 6, we can see the different energy consumption structures among sectors, and coal is mainly consumed in the power and industry sectors. Only under the SSP1-26 scenario is the future's total consumption of coal decreased compared to 2015. By 2050, the total coal consumption could reach nearly 6 billion tonnes by 2050 under the SSP5-85 scenario with an ∼ 60 % increase during 2015–2050. Under the SSP3-70, SSP4-60, and SSP5-85 scenarios, the increase in coal consumption mainly occurs before 2030, and a slight decrease in coal consumption occurs during 2030–2050, except in the SSP5-85 scenario. In contrast, the decrease in coal consumption occurs during 2030–2050 under the SSP1-26

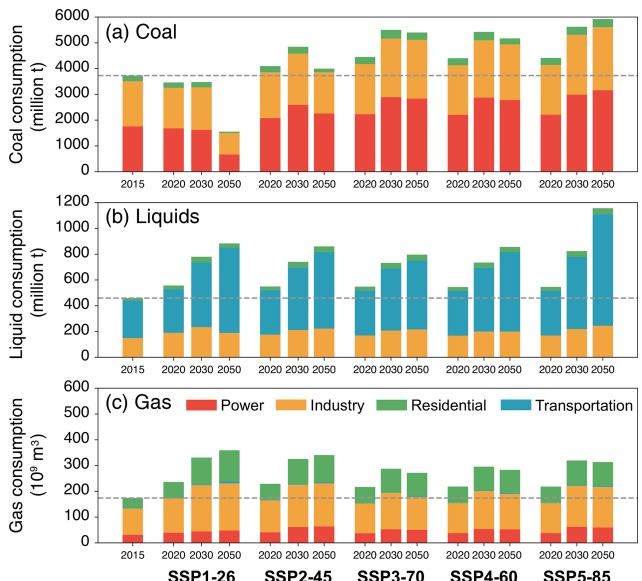

**Figure 6.** China's future energy consumption in the years 2020, 2030, and 2050 under five energy scenarios. The fuel types plotted here include **(a)** coal, **(b)** liquids, and **(c)** gas. Energy consumption is divided into four energy-related sectors (stacked column chart): power, industry, residential, and transportation.

scenario, which implies the acceleration of energy transformations in the far future to meet the stringent climate target. Liquids are mainly consumed in the transportation sector (63.5 % in 2015), and most of the rest is consumed in the industry sector mainly for feedstock (31.9 % of the total).

Liquid consumption shows a significant increase in the transportation sector even under the SSP1-26 scenario due to the ever-increasing vehicle demand and limited fuel switching considered in the GCAM-China model (Fig. S7). Thus, 16.3 %, 71.7 %, and 126.6 % increases are achieved in the transportation sector in 2020, 2030, and 2050, respectively, under the SSP1-26 scenario compared to 2015. The liquid consumption in the industrial sector is relatively stable with small changes. The growth rates of total liquid consumption slow down during 2030–2050 under all energy scenarios except the SSSP5-85 scenario, which is mainly driven by changes in liquid consumption in the transportation sector. Gas is mainly consumed in the industrial, residential, and power sectors, accounting for 58.2 %, 23.1 %, and 18.1 %, respectively, of the total consumption in 2015. In the future, more gas would be consumed under the lower global warming target because gas is defined as a clean fossil fuel compared to coal and liquids. The increase in gas consumption mainly occurs in the residential sector under all energy scenarios driven by the ever-increasing demand and energy policy of replacing coal with gas in the future's residential energy structure. Under the most stringent climate target, gas consumption increased by 199.4 % from 2015 to 2050 compared to 54.1 % in the power sector and 79.7 % in the indus-

try sector. In total, the more stringent the climate target is, the larger the required adjustments in the future energy structure.

## 4.2 Emission trends during 2010–2050

Figure 7 shows the historical and future emission trends of major air pollutant emissions ($SO_2$, $NO_x$, $PM_{2.5}$ and NMVOCs) from 2010 to 2050 under six designed scenarios. Historical emissions during 2010–2015 are obtained from MEIC, and all designed scenarios represent different emission mitigation pathways under different evolution of future societal conditions and climate and environmental policies. China's historical anthropogenic emissions during 2010–2015 are estimated to peak in 2011, 2012, and 2011 for $SO_2$, $NO_x$, and $PM_{2.5}$, respectively. In addition, $SO_2$, $NO_x$, and $PM_{2.5}$ emissions declined to 17.4, 23.7, and 9.1 Tg by 2015 (Table 2), respectively, which mainly resulted from a series of effective environmental policies applied over the past few years. In contrast, NMVOC emissions have persistently increased by 17.2 % from 2010 to 2015.

In the SSP3-70-BAU and SSP4-60-BAU scenarios, under the pessimistic development trends with limited investments and attention to climate and environmental issues in China, the emissions of major air pollutants would slightly change except for an obvious increase in $NO_x$ emissions (∼ 43 % CE6 in both the SSP3-70-BAU and SSP4-60-BAU scenarios) in the next 35 years. Due to the continued effect of the Action Plan and other current environmental policies, there are still obvious decreases in emissions under the SSP4-60-BAU scenario in the next few years, with 12.9 % of $SO_2$, 5.8 % of $NO_x$, and 4.7 % of $PM_{2.5}$ emissions decreasing from 2015 to 2018. This result implies that stringent and effective environmental policies are needed and play an important role in medium and long-term emission mitigation. Although there is a relatively lax radiative forcing target in the SSP5-85-BHE scenario, all the major air pollutant emissions are largely reduced by enhanced emission control measures when comparing the SSP5-85-BHE scenario with the base year 2015, especially during 2015–2030. On the contrary, similar energy structure but lax pollution controls drive the stable or increasing emissions under the SSP3-70-BAU scenario. In 2030, $SO_2$, $NO_x$, $PM_{2.5}$, and NMVOC emissions under the SSP5-85-BHE scenario further decrease by 68 %, 67 %, 59 %, and 48 % CE7 compared to the SSP3-70-BAU scenario, respectively. Meanwhile, low-carbon energy development also plays an equally important role in future emission mitigation. The SSP1-26-ECP scenario could further reduce the $SO_2$, $NO_x$, $PM_{2.5}$, and NMVOC emissions by 24 %, 17 %, 13 %, and 3 % in 2030 compared to the SSP2-45-ECP scenario due to respective low-carbon energy transitions during 2015–2030. In addition, low-carbon energy transitions would have much larger benefits in the far future, and the $SO_2$, $NO_x$, $PM_{2.5}$, and NMVOC emissions under the SSP1-26-ECP scenario would be 52 %, 34 % CE8, 30 %, and 5 %

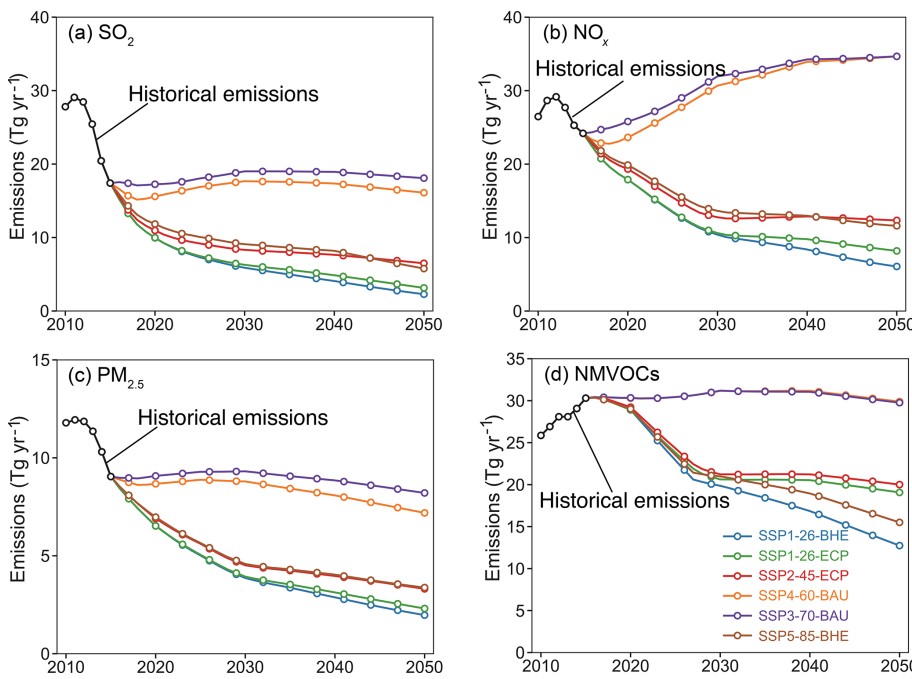

**Figure 7.** Emissions of major air pollutants in China from 2010 to 2050. The species plotted here include **(a)** $SO_2$, **(b)** $NO_x$, **(c)** $PM_{2.5}$, and **(d)** NMVOCs. This figure shows the historical annual emission data from 2010 to 2015 and emission projections (one dot every 3 years) under six designed scenarios during 2015–2050 (SSP1-26-ECP, SSP1-26-BHE, SSP2-45-ECP, SSP3-70-BAU, SSP4-60-BAU, and SSP5-85-BHE).

lower, respectively, than those under the SSP2-45-ECP scenario in 2050.

In addition, under the same socio-economic and energy pathways, the SSP1-26-ECP and SSP1-26-BHE scenarios have similar emission mitigation pathways during 2015–2030 due to similar and strict enforcement of environmental policies, while the SSP1-26-BHE scenario can further reduce $SO_2$, $NO_x$, $PM_{2.5}$, and NMVOC emissions by 27 %, 26 %, 15 %, and 33 % CE9, respectively, by 2050 after applying the best-available technologies compared to the SSP1-26-ECP scenario. However, we found that limited mitigation is achieved except for NMVOC emissions during 2030–2050 from applying the best-available technologies compared to average emission decrease rates between 2015–2030 and 2030–2050. This implies that the emission mitigation potential from emission control measures is gradually exhausted in the long-term actions. Compared to the emission control measures, low-carbon transitions play a more substantial role in the medium and long-term mitigation pathways.

Different sectors have different emission reduction potential and mitigation pathways under various scenarios. Figure 8 further shows the sectoral emission contributions of major air pollutants under all designed scenarios in the years 2020, 2030, and 2050. As shown in Fig. 8, the most important sector identified in 2015 is the industrial sector for all major air pollutants, which contributes 59 %, 41 %, 48 %, and 33 % of $SO_2$, $NO_x$, $PM_{2.5}$, and NMVOC emissions, respectively.

For $SO_2$ emissions, the industrial sector mainly drives the changes under all the scenarios; for instance, almost 56 %–59 % of the total emission reductions are obtained from the industrial sector in 2050 under the SSP5-85-BHE, SSP2-45-ECP, SSP1-26-BHE, and SSP1-26-ECP scenarios. Driven by the ultra-low emission standard, $SO_2$ emissions from the power sector rapidly decrease in the near future (i.e. 2015–2020), which is similar to the $NO_x$ and $PM_{2.5}$ emissions. In addition, the $SO_2$ emissions from the residential sector largely decrease through low-carbon transitions when comparing the SSP1-26-BHE and SSP5-85-BHE scenarios with the base year 2015. For $NO_x$, emissions from all the sectors increase under the SSP3-70-BAU and SSP4-60-BAU scenarios due to no additional environmental policies being applied during 2015–2050. $NO_x$ emissions under the SSP5-85-BHE scenario are largely reduced, while the SSP1-26-BHE scenario has relatively large but similar reductions for the industrial and transportation sectors in 2030, which implies that low-carbon energy transitions have limited effects on $NO_x$ emission reductions during 2015–2030. In 2050, low-carbon energy transitions could further reduce $NO_x$ emissions from the power and industrial sectors by 78 % and 50 % through power structure and industrial structure adjustments.

The reductions in $PM_{2.5}$ emissions under the SSP3-70-BAU and SSP4-60-BAU scenarios are contributed by the residential sector through the wide application of advanced residential stoves. Policies on industrial structure adjustment

**Table 2.** Anthropogenic emissions of air pollutants in 2015, 2030, and 2050 under different scenarios (unit: $Tg\,yr^{-1}$).

| Year | $SO_2$ | $NO_x$ | NMVOCs | $PM_{2.5}$ | $PM_{10}$ | CO | BC | OC | $NH_3$ | $CO_2$ / 1000 |
|---|---|---|---|---|---|---|---|---|---|---|
| 2015 | 17.4 | 23.7 | 30.3 | 9.1 | 23.7 | 153.6 | 1.5 | 2.6 | 10.5 | 10.5 |
| **SSP5-85-BHE** | | | | | | | | | | |
| 2030 | 9.1 | 13.6 | 21.0 | 4.6 | 6.2 | 121.8 | 0.6 | 1.2 | 8.4 | 14.1 |
| 2050 | 5.8 | 11.6 | 15.5 | 3.4 | 4.4 | 122.8 | 0.4 | 0.7 | 6.0 | 16.9 |
| (2030–2015) / 2015 | −47.8 % | −42.5 % | −30.7 % | −49.9 % | −49.8 % | −20.7 % | −61.1 % | −53.2 % | −20.0 % | 34.2 % |
| (2050–2030) / 2030 | −36.5 % | −14.9 % | −26.1 % | −26.3 % | −29.1 % | 0.8 % | −32.9 % | −41.8 % | −28.1 % | 19.8 % |
| (2050–2015) / 2015 | −66.7 % | −51.1 % | −48.8 % | −62.6 % | −81.4 % | −20.1 % | −73.3 % | −73.1 % | −42.9 % | 61.0 % |
| **SSP3-70-BAU** | | | | | | | | | | |
| 2030 | 19.0 | 31.9 | 31.2 | 9.3 | 12.5 | 159.6 | 1.4 | 2.3 | 10.8 | 15.2 |
| 2050 | 18.1 | 34.7 | 29.8 | 8.2 | 11.2 | 157.1 | 1.1 | 1.8 | 10.0 | 16.2 |
| (2030–2015) / 2015 | 9.2 % | 34.8 % | 3.0 % | 1.9 % | 1.8 % | 3.9 % | −6.3 % | −9.2 % | 2.8 % | 45.2 % |
| (2050–2030) / 2030 | −4.9 % | 8.5 % | −4.6 % | −11.8 % | −10.9 % | −1.6 % | −18.3 % | −22.8 % | −6.8 % | 6.5 % |
| (2050–2015) / 2015 | 4.0 % | 46.4 % | −1.7 % | −9.9 % | −52.7 % | 2.3 % | −26.7 % | −30.8 % | −4.8 % | 54.3 % |
| **SSP4-60-BAU** | | | | | | | | | | |
| 2030 | 17.7 | 30.7 | 31.2 | 8.8 | 11.8 | 155.3 | 1.3 | 2.3 | 10.8 | 13.2 |
| 2050 | 16.1 | 34.7 | 29.9 | 7.2 | 9.8 | 144.4 | 1.2 | 1.5 | 10.0 | 11.8 |
| (2030–2015) / 2015 | 1.5 % | 29.4 % | 2.9 % | −3.8 % | −3.9 % | 1.1 % | −7.8 % | −11.4 % | 2.8 % | 26.1 % |
| (2050–2030) / 2030 | −8.8 % | 13.0 % | −4.1 % | −18.2 % | −17.3 % | −7.0 % | −19.2 % | −34.5 % | −7.1 % | −10.7 % |
| (2050–2015) / 2015 | −7.5 % | 46.4 % | −1.3 % | −20.9 % | −58.7 % | −6.0 % | −20.0 % | −42.3 % | −4.8 % | 12.4 % |
| **SSP2-45-ECP** | | | | | | | | | | |
| 2030 | 8.3 | 12.8 | 21.2 | 4.5 | 5.9 | 109.4 | 0.5 | 1.3 | 8.7 | 11.9 |
| 2050 | 6.5 | 12.3 | 20.0 | 3.3 | 4.3 | 107.0 | 0.4 | 0.9 | 7.9 | 8.9 |
| (2030–2015) / 2015 | −52.2 % | −46.1 % | −30.0 % | −50.6 % | −52.0 % | −28.8 % | −62.4 % | −51.0 % | −16.8 % | 13.2 % |
| (2050–2030) / 2030 | −22.0 % | −3.4 % | −5.7 % | −26.6 % | −27.1 % | −2.2 % | −28.0 % | −31.8 % | −9.9 % | −25.2 % |
| (2050–2015) / 2015 | −62.6 % | −48.1 % | −34.0 % | −63.7 % | −81.9 % | −30.3 % | −73.3 % | −65.4 % | −24.8 % | −15.2 % |
| **SSP1-26-ECP** | | | | | | | | | | |
| 2030 | 6.3 | 10.6 | 20.6 | 3.9 | 5.0 | 91.1 | 0.5 | 1.2 | 8.7 | 9.7 |
| 2050 | 3.1 | 8.2 | 19.1 | 2.3 | 2.7 | 74.4 | 0.3 | 0.8 | 7.9 | 4.1 |
| (2030–2015) / 2015 | −63.9 % | −55.2 % | −31.9 % | −56.9 % | −59.8 % | −40.7 % | −65.8 % | −54.3 % | −16.8 % | −9.4 % |
| (2050–2030) / 2030 | −50.0 % | −22.9 % | −7.5 % | −41.4 % | −44.6 % | −18.4 % | −30.4 % | −35.7 % | −9.9 % | −57.3 % |
| (2050–2015) / 2015 | −82.2 % | −65.4 % | −37.0 % | −74.7 % | −88.6 % | −51.6 % | −80.0 % | −69.2 % | −24.8 % | −61.0 % |
| **SSP1-26-BHE** | | | | | | | | | | |
| 2030 | 5.9 | 10.4 | 19.9 | 3.9 | 4.9 | 91.0 | 0.5 | 1.2 | 8.5 | 9.7 |
| 2050 | 2.3 | 6.1 | 12.7 | 2.0 | 2.3 | 69.5 | 0.3 | 0.7 | 6.1 | 4.1 |
| (2030–2015) / 2015 | −66.2 % | −56.2 % | −34.4 % | −57.7 % | −60.6 % | −40.7 % | −66.2 % | −54.9 % | −19.2 % | −9.4 % |
| (2050–2030) / 2030 | −61.2 % | −41.6 % | −35.9 % | −49.0 % | −53.0 % | −23.6 % | −43.0 % | −42.7 % | −27.9 % | −57.3 % |
| (2050–2015) / 2015 | −86.8 % | −74.3 % | −58.1 % | −78.0 % | −90.3 % | −54.8 % | −80.0 % | −73.1 % | −41.9 % | −61.0 % |

can lead to $PM_{2.5}$ emission reductions of 78 % from 2015 to 2050 under the SSP1-26-ECP scenario, and the best-available technologies could further reduce 14 % of industrial emissions by 2050 when comparing the SSP1-26-ECP and SSP1-26-BHE scenarios. Low-carbon energy transitions have very limited effects on $PM_{2.5}$ emission reductions from the residential sector when comparing the SSP1-26-BHE and SSP5-85-BHE scenarios because biomass dominates the $PM_{2.5}$ emissions and there are similar projections of biomass consumption under these two energy scenarios. NMVOC emissions are dominated by the solvent use and industrial sectors in 2015. Under the SSP3-7-BAU and SSP4-60-BAU

scenarios, NMVOC emissions changed slightly during 2015–2050, mainly due to emission increases from the industrial sector, which partly offsets the decrease from the transportation sector. Under the SSP5-85-BHE scenario, emission reductions from the industrial and solvent use sectors contribute 23 % and 32 % of total NMVOC reductions, respectively. NMVOC emissions from the industrial and solvent use sectors are only reduced by 58 % and 43 %, respectively, under the SSP1-26-BHE scenario during 2015–2050. Even under the SSP1-26-ECP scenario, limited NMVOC reductions from industry and solvent use sectors are obtained, which implies that controlling NMVOC emissions in the future is chal-

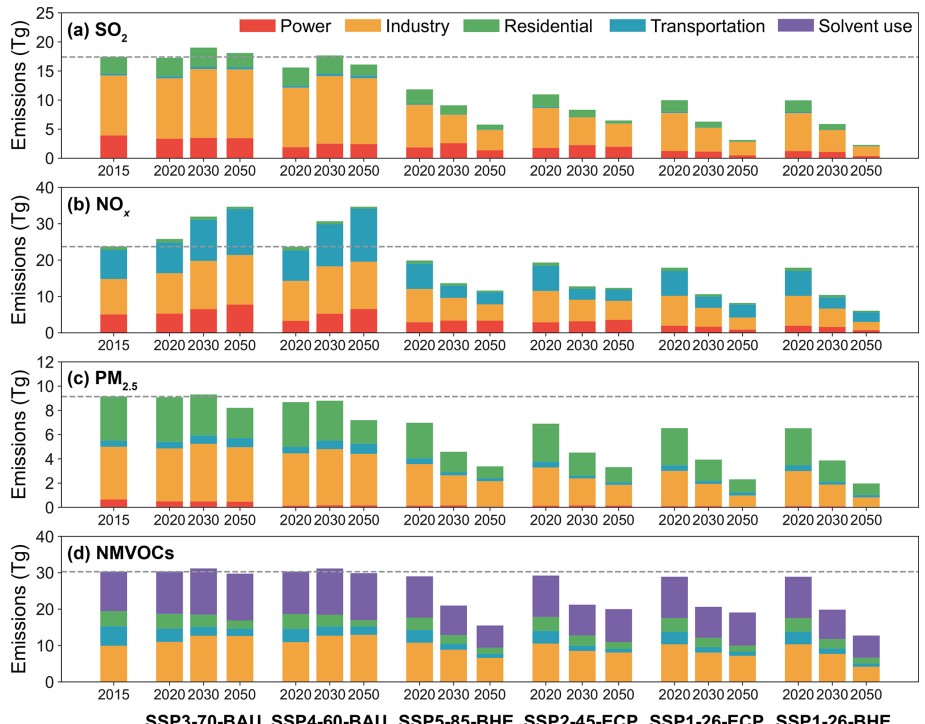

**Figure 8.** China's future anthropogenic emissions by sector in the years 2020, 2030, and 2050 under six scenarios. The species plotted here include **(a)** $SO_2$, **(b)** $NO_x$, **(c)** $PM_{2.5}$, and **(d)** NMVOCs. Emissions are divided into five source sectors (stacked column chart): power, industry, residential, transportation, and solvent use.

lenging compared to other major air pollutants, particularly in the industrial and solvent use sectors. By 2050, if low-carbon energy transitions and best-available technologies are fully achieved, under the SSP1-26-BHE scenario, the industrial sector is still the dominant emission source for all the major air pollutants, in addition to the transportation sector for $NO_x$, the residential sector for $PM_{2.5}$, and the solvent use sector for NMVOC emissions. Therefore, we distinguished the key emission sources during different periods, and the control measures should be strengthened in the future.

## 5 Discussion

### 5.1 Comparison with emission estimates from the harmonized CMIP6 emissions dataset

In this study, although we created our scenarios based on the CMIP6 global development modes and societal conditions, more realistic short- and long-term emission control policies are integrated into our emission scenarios in China. Here, we compare the emissions under corresponding scenarios from our study and the harmonized CMIP6 emissions dataset (Fig. 9; Gidden et al., 2019). There are obvious gaps for major air pollutant emissions in the base year except for NMVOCs, and the $SO_2$, $NO_x$, and black carbon (BC) emissions in 2015 from the CMIP6 database are

higher than those from our MEIC emission inventory by 43 %, 29 %, and 79 %, respectively. These gaps are mainly caused by the underestimation of emission reductions obtained from China's Action Plan in the CMIP6 database, and the emission bias in the base year would pass to the future and lead to different emission mitigation pathways. Other than the emission gaps in the base year, there are also obvious differences in future emission trends under corresponding scenarios. In the CMIP6 database, emissions under the SS3-70-weak and SSP4-60-weak scenarios show opposing future trends, while emissions under the SSP3-70-BAU and SSP4-60-BAU scenarios from this work have similar trends and slight changes except for the $NO_x$ emissions during 2015–2050. In the CMIP6 estimates, the projections of near-term emission factor (EF) evolution are mainly based on current policies and technological options derived from the GAINS model, while long-term EF evolution for weak and strong pollution control scenarios has employed different assumptions among different IAMs (Rao et al., 2017). Our estimates of near- and long-term EF evolution are both driven by environmental policies based on the same framework. Therefore, the continuous effects of the Action Plan could not offset the ever-increasing energy demand during 2015–2030; consequently, emissions such as $SO_2$ and $NO_x$ increase in the corresponding period. The SSP5-85-BHE and SSP5-85-strong scenarios have similar mitigation trends ex-

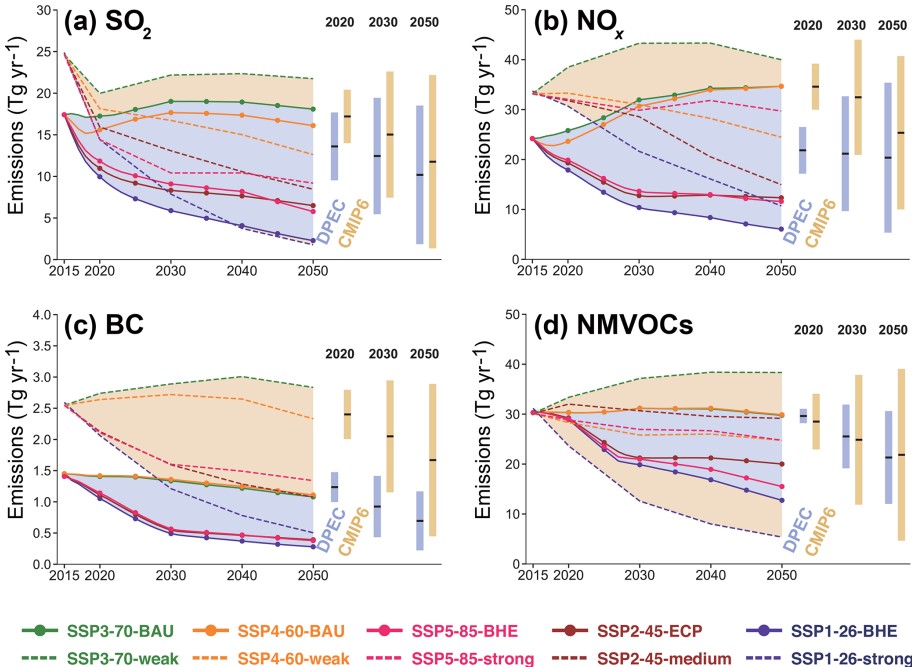

**Figure 9.** Comparison of future emissions estimated in this study with estimates from the harmonized CMIP6 emissions dataset. The species plotted here include **(a)** $SO_2$, **(b)** $NO_x$, **(c)** BC, and **(d)** NMVOCs. The scenarios include SSP1-26-BHE, SSP2-45-ECP-SSP3-70-BAU, SSP4-60-BAU, and SSP5-85-BHE scenarios from this study and the corresponding SSP1-26-strong, SSP2-45-medium, SSP3-70-weak, SSP4-60-weak, and SSP5-85-strong scenarios from the harmonized CMIP6 emissions dataset.

cept for NMVOC emissions, and larger reductions are obtained before 2030. The NMVOC emissions under the SSP5-85-strong scenario in the CMIP6 are probability driven by the assumptions of a high economic pace and few technology options. While under the SSP5-85-BHE scenario, we still assumed that the best-available technologies are fully applied. When comparing the SSP2-45-ECP and SSP2-45-medium scenarios, we see that similar emission reduction trends occurred during 2015–2030, except for the NMVOC emissions, while continuous emission reductions were observed under the SSP2-45-medium scenario, which is mainly caused by different policy assumptions. Thus, we assumed that no environmental policies are further considered after achieving national air quality standards by 2030. Under the most optimistic development modes and strict forcing targets (i.e. the SSP1-26-BHE and SSP1-26-strong scenarios), all major air pollutant emissions rapidly decrease, while emissions in the year 2050 among species show various differences. $SO_2$ and BC emissions have relatively small differences compared to $NO_x$ and NMVOC emissions between these two scenarios. CMIP6 has more optimist projections for NMVOC emission reductions; in fact, the difficulty in reducing NMVOC emissions is much larger than for fossil-fuel-dominated species because the emission sources are highly dispersive. Therefore, we projected a limited NMVOC emission reduction even with the best-available technologies applied. In summary, the emission difference ranges of various species be-

tween two sets of scenarios are much larger in the near future, for example 2020, while the ranges are gradually expanded over time, and eventually our scenarios have smaller differences with the CMIP6 scenarios. This result is mainly because our scenarios are based on the more realistic Chinese development in the near future under the constraint of issued environmental policies and emissions in the base year.

Additionally, we compared the $CO_2$ emissions between two sets of scenarios, as shown in Fig. 10, and the corresponding scenarios between the CMIP6 scenarios and ours are quite similar due to the same energy scenarios being adopted. The differences are mainly from the different $CO_2$ EFs adopted. Figure 10 shows that $CO_2$ emissions would continue increasing until 2050 when radiative forcing targets are above $7.0 \, \text{W m}^{-2}$. Under the SSP3-70-BAU and SSP5-85-BHE scenarios, $CO_2$ emissions increase by 55 % and 61 % during 2015–2050, respectively. Under the radiative forcing targets of 6.0 and $4.5 \, \text{W m}^{-2}$, $CO_2$ emissions begin to decrease during 2030–2035, and finally emissions in 2050 under the SSP2-45-ECP scenario would be lower than the emission levels of 2015. For the SSP1-26-BHE scenario, $CO_2$ emissions would start to decrease in the very near future (during 2015–2020), and a 61 % reduction is achieved during 2015–2050.

In particular, we compare the sectoral emissions under the SSP1-26-BHE scenario from this work and the SSP1-26-strong scenario from the CMIP6 database, and the sector

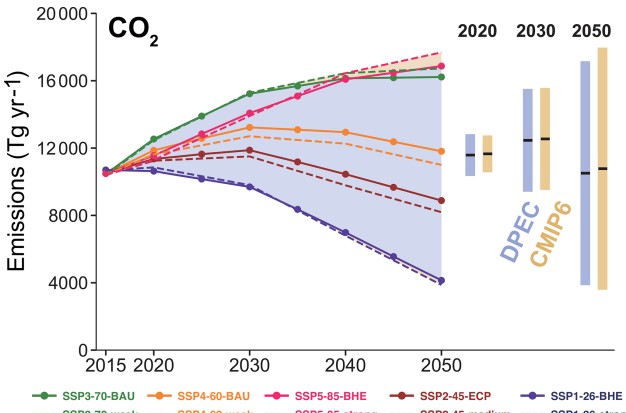

**Figure 10.** Comparison of future $CO_2$ emissions estimated in this study with estimates from the harmonized CMIP6 emissions dataset. The scenarios include SSP1-26-BHE, SSP2-45-ECP, SSP3-70-BAU, SSP4-60-BAU, and SSP5-85-BHE scenarios from this study and the corresponding SSP1-26-strong, SSP2-45-medium, SSP3-70-weak, SSP4-60-weak, and SSP5-85-strong scenarios from the harmonized CMIP6 emissions dataset.

maps are shown in Table S4. As shown in Fig. 11, the differences in the base year are mainly contributed by industry for the $SO_2$ and $NO_x$ emissions and the power and heating sector for NMVOC and BC emissions. This is probability because of the underestimations in the $SO_2$ and $NO_x$ control levels in the industrial sector and different EFs chosen for the NMVOCs and BC in the power and heating sector. During 2015–2050, the $SO_2$ and $NO_x$ emissions under the SSP1-26-strong scenario gradually closed the gaps with the SSP1-26-BHE scenario. For the NMVOC emissions, the difference in 2050 is caused by different projections in the industrial and solvent use sectors. Under the projections from the SSP1-26-strong scenario, the emission reductions for NMVOCs during 2015–2030 are dominated by the industrial and power and heating sectors. During 2030–2050, the emission reductions are dominated by the solvent use sector. For BC emissions, in addition to the difference from the power and heating sector, the difference between these two scenarios is magnified by the industrial sector, and the CMIP6 has limited emission reduction from the industrial sector, which becomes the dominant sector by 2050. Under the SSP1-26-BHE scenario, except for the dominant contributions from the residential sector during 2015–2050, the transportation sector became another dominant sector by 2050 due to limited emission reductions.

## 5.2 Limitations and uncertainties

In this study, a dynamic emission projection model was developed to estimate the evolution of future air pollutants and $CO_2$ emissions, and a comprehensive understanding of future emission trends was achieved by connecting various socio-economic developments and climate targets and differ-

ent pollution control policies during 2015–2050. Under the strictest scenario designed in this study (i.e. SSP1-26-BHE scenario), from 2015 to 2050, emission reductions of 87 % for $SO_2$, 74 % for $NO_x$, 78 % for $PM_{2.5}$, and 55 % CE10 for NMVOCs could be achieved under a combination of low-carbon energy transitions and best-available environmental policies. During 2015–2030, end-of-pipe controls play a very important role in future air pollutant emission mitigation to facilitate staged air quality targets when comparing the SSP5-85-BHE and SSP3-70-BAU scenarios. With the exhaustion of the emission reduction potential from end-of-pipe controls, low-carbon energy transitions could not only reduce $CO_2$ emissions to meet the climate target but also fundamentally reduce the rest of emissions during 2030–2050 to achieve long-term air quality targets when we compared the SSP5-85-BHE and SSP1-26-BHE scenarios. Our analyses identify the feasible pathways for achieving emission reductions for air pollutants to maximally protect public health. In fact, the flexibility and compatibility of the DPEC can be applied in the pre-evaluations and post-evaluations of various future policies and regulations. More importantly, DPEC runs in 1-year time steps starting from the base year and moving into the future, and these runs allow for tracking of the annual effectiveness of each policy, which is especially useful for short-term evaluations.

There are several limitations and uncertainties in this study. First, the energy scenarios we used in this work are derived from the GCAM-China model, in which the reference scenario is counterfactual and does not explicitly consider mitigation actions. For instance, China aims to develop renewable energy in the power sector to create clean electricity in the future. The effects of clean energy power are expected to increase rapidly in the future. As announced in the 13th FYP, the generation share of renewable energy is planned to increase to 27 % by 2020. Even the SSP1-26 scenario underestimates the actions taken on the adjustment of power energy structure by the Chinese government (20 % of renewable energy in 2020; Fig. S8). Additionally, China has launched several initiatives to promote electric vehicles and aims to increase the number of electric vehicles to 5 million in 2020 according to the development plan for new-energy vehicles (Wang et al., 2014). In contrast, all the energy scenarios obtained from the GCAM-China model except the SSP1-26 scenario have low projections of the future effects of electric vehicles (Fig. S7), and the effects of new-energy vehicles only increase to ∼ 2 % by the end of 2050. This penetration result is mainly because the assumption of high electric vehicle costs in the GCAM-China model we used. It is proven that increased vehicle electrification has a net positive impact on air quality, climate change, and human health (Liang et al., 2019). In the future, a long-term energy scenario coupled with China's short and long-term energy policies is needed to accurately estimate the future mitigation potential or project future emission mitigation pathways.

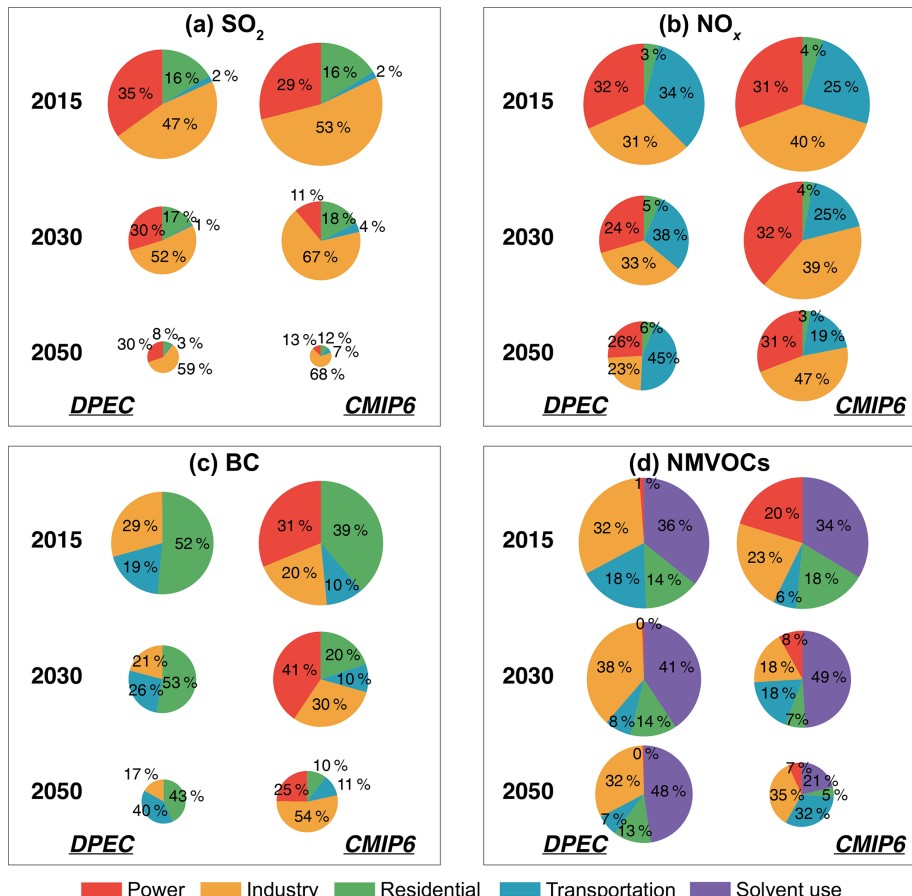

**Figure 11.** The comparison of sectoral emissions between the SSP1-26-BHE scenario from the DPEC from this study and SSP1-26-strong scenario from the harmonized CMIP6 emissions dataset. **(a)** SO$_2$, **(b)** NO$_x$, **(c)** BC, and **(d)** NMVOCs emissions. For each pollutant, the relative change in the radius of the pie chart is proportional to the change in emissions.

Secondly, the policies promulgated from governments usually only have macro measures and completion years without yearly detailed and parameterized actions. Our parameterized process within each scenario may underestimate or overestimate the emission reductions from each measure. For example, the effectiveness of measures targeting small and scattered emission sources (e.g. phasing out small and old industrial factories and eliminating small coal-fired industrial boilers) is difficult to evaluate and reasonably parameterize, which may lead to higher uncertainty ranges in future emission estimates.

Thirdly, the CMIP6 dataset we applied to compare is from different IAMs. Both assumptions and models would impact the results among different scenarios, but we only considered the impacts of scenario assumptions and included policies in our study. Future studies should focus on the discrepancies and be led by different IAMs, including design sensitivity simulations with fixed IAMs to quantify the uncertainties.

Finally, emission estimates in the base year are uncertain due to incomplete knowledge of underlying data (Zhao et al., 2011; Liu et al., 2015). The uncertainties from the historical

emission inventories are widely quantified in previous works (e.g. Zhang et al., 2009; Lei et al., 2011a; Lu et al., 2011; Li et al., 2017). These uncertainties may pass to our projection model and create new uncertainties in the emission reduction rates and future emission mitigation pathways, but there are few impacts on the emission estimates for the year 2050.

## 5.3 Policy implications

Air quality improvement and climate change governance are of equal importance in future environmental management for China. Both air pollution and climate change issues are essentially energy problems, especially coal problems in the current state of China. On the one hand, actions of energy conservation and low-carbon energy transitions to reduce CO$_2$ emissions also reduce co-emitted air pollutants such as SO$_2$, NO$_x$ and PM$_{2.5}$, creating co-benefits for air quality. Therefore, in this study, we emphasize the importance of air pollution and climate co-governance under the current environmental situation for China. The Chinese government should strengthen the co-governance policy design in the fu-

ture to achieve maximal co-benefit effects with the least action and investment. On the other hand, we found that active clean air policies in China could reduce near-term air pollutant emissions more significantly. In contrast, limited air quality improvement could be obtained from low-carbon energy transitions in the near future due to the inertia of current energy systems and no quick switch to low-carbon energy (Kramer et al., 2009; Tong et al., 2019). Therefore, the quick promotion of ultra-low emission standards in the power and industrial sectors, as well as the relatively low emission standards in other sectors, is vital for meeting near-term air quality targets. In summary, in this study, employing the sophisticated dynamic model framework we constructed by linking a global energy system model to a regional emission inventory model, we conducted a comprehensive assessment from the air quality and climate co-governance perspectives through scenario analysis, which provides important insights into the impacts of China's future emissions on global climate and environment change as well as future air quality and climate co-governance in China. Our developed scenarios can offer a better understanding of future trends in air pollution and greenhouse gas (GHG) emissions and changes in atmospheric composition over China under a range of IPCC AR6 global socio-economic and climate scenarios and local air pollution polices. In the future, we will continue assessing more future emission pathways for policymakers on the basis of this framework.

*Data availability.* Emission data (China's future emission scenario and database 2015–2050) generated from this study are available at http://www.meicmodel.org/dataset-dpec.html (last access: 23 April 2020).

*Supplement.* The supplement related to this article is available online at: https://doi.org/10.5194/acp-20-1-2020-supplement.

*Author contributions.* QZ designed the research; DT, JC, YL, LY, CH, YQ, HZ, and YZ developed the emission projection model; SY and LC developed the GCAM-China model; ML, FL, and BZ provided historical emission data; QZ, DT, JC, YL, SY, GG, and LC developed future emission scenarios and interpreted data; DT, JC, and QZ prepared the manuscript with contributions from all co-authors.

*Competing interests.* The authors declare that they have no conflict of interest.

*Acknowledgements.* This work was supported by the National Key R&D programme and the National Natural Science Foundation of China. We thank the Energy Foundation China for financial support. Sha Yu was supported by the Global Technology Strategy Project (GTSP).

*Financial support.* This research has been supported by the National Key R&D program (grant no. 2016YFC0208801), the National Natural Science Foundation of China (grant no. 91744310, 41921005, and 41625020), the Energy Foundation China (G-1806-28044), and the National Research Program for key issues in air pollution control (DQGG0201).

*Review statement.* This paper was edited by Aijun Ding and reviewed by two anonymous referees.

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

**Remarks from the language copy-editor**

CE1   It is our house standard to show plurality in abbreviations. This has been made consistent throughout and will stay as is.

CE2   An en dash means "and". This will not be changed here or elsewhere, unless the meaning is incorrect.

CE3   Do you mean "industrial solvent use"?

CE4   FW is not used elsewhere in the paper so the abbreviation was not inserted here.

CE5   The standard SI abbreviation is kW h.

CE6   Please give an explanation of why this needs to be changed. We have to ask the handling editor for approval. Thanks.

CE7   Please give an explanation of why this needs to be changed. We have to ask the handling editor for approval. Thanks.

CE8   Please give an explanation of why this needs to be changed. We have to ask the handling editor for approval. Thanks.

CE9   Please give an explanation of why this needs to be changed. We have to ask the handling editor for approval. Thanks.

CE10   Please give an explanation of why this needs to be changed. We have to ask the handling editor for approval. Thanks.

**Remarks from the typesetter**

TS1   Please explain what needs to be corrected within the figures. For some cases, it is easier for us to apply the correction to the existing figure file rather than replacing it.