# Peer review of "Dynamic projection of anthropogenic emissions in China: methodology and 2015-2050 emission pathways under a range of socioeconomic, climate policy, and pollution control scenarios"

_Atmospheric Chemistry and Physics, 2019_

## Referee Comment (RC1) · Anonymous Referee #1 · 30 Jan 2020

This work aims to develop a dynamic projection model connecting SSP-RCP scenarios with local policies and representing dynamic emission changes under local policies for China, so as to explore China's future anthropogenic emission pathways. I think this work is of great importance since that most of China's local emission-control policies and technologies have not been well incorporated in global emission scenarios yet, which may introduce great uncertainties in climate and socioeconomic assessment. This work can substantially fill such gaps. Generally, this manuscript is well-written and well-structured, and fits the scope of this journal. Here are some issues that are

suggested to be addressed for further improving this work.

In Section 3.1, five SSP-RCP combinations were chosen, please briefly discuss the reasons and designs for the combination options, like choosing SSP3-60 and SSP4-70 rather than SSP3-70 and SSP4-60.

Since the GCAM-China energy outputs were harmonized by multiplying a balance ratio to eliminate the discrepancy between MEIC and GCAM-China models in the base year. I am concerned about the spatial heterogeneity for the balance ratio during 2010-2015. Could a similar harmonization method be applied for the gap in the base year for CMIP6 and DPEC?

Minor comments:

Page2, Line 53: 'is still higher than'

Page2, Line 58: 'benefits substantial reductions', improper use of verb 'benefit'

Page3, Line 74-75: 'These scenarios describe future emission 75 changes based on a set of scenarios', better to differentiate the 'scenarios' in expression

Page3, Line 78: simplify 'do not consider' to 'neglect'?

Page3, Line 80-81: unclear expression 'The absence of public emissions data products makes further use difficult in climate and chemical transport models.'

Page3, Line 82-83: confusing expression 'that connects global scenarios considering local policies and representing dynamic emission changes with local policies and technology deployment', rewrite as 'a comprehensive scenario set connecting SSP-RCP scenarios with local policies and representing dynamic emission changes under local policies'?

Page4, Line 96-98: please rephrase this sentence to make it clear

Page5, Line 145; Page13, Line 376-377, 394; Page15, Line 449-452: 'Therefore, we

used. . .', better to keep consistency in tense (past tense for methodology)

Page8, Line 219: 'A technology-based emission projection model', remove 's'

Page11, Line 315: 'GCAM-China'

Page12, Line 340 and 349; Page13, Line 391; Page18, Line 542: please pay attention to tense for the clauses of 'we assumed that', 'we found that'

Page15, Line 435: try to use proper pronoun to make sentence brief

Page18, Line 419: 'is required to be up to'

---

## Referee Comment (RC2) · Anonymous Referee #2 · 1 Mar 2020

Due to its substantial energy demand and huge population, China's future air pollutants and greenhouse gas emissions are of great importance both locally and internationally. This work developed a dynamic projection model to predict future emissions under various SSP-RCP scenarios, with a particular focus to integrate local policies. This is an important contribution, and will help to fill the missing gap from those global-scale studies. I suggest minor revisions for acceptance.

1. SSP and RCP scenarios can interact in different combinations. I understand it is impractical to simulate all scenario combinations, but maybe the author could consider adding some justifications of why choosing SSP1-26, SSP2-45, SSP3-70, SSP4-60, and SSP5-85 scenarios? Also, it would be could to add a few sentences about the implications due to such choices?

[Figure]

2. The author compared their predicted emissions with CMIP6 results, which reveal notable differences. Could the author provide some relevant guidance/instructions for CMIP6 and future DPEC users? For instance, which model is more reliable/useful under which circumstances? How to interpret the results from these different methods, respectively, in China's context?

---

## Author Comment (AC1) · 5 Mar 2020

This work aims to develop a dynamic projection model connecting SSP-RCP scenarios with local policies and representing dynamic emission changes under local policies for China, so as to explore China's future anthropogenic emission pathways. I think this work is of great importance since that most of China's local emission-control policies and technologies have not been well incorporated in global emission scenarios yet, which may introduce great uncertainties in climate and socioeconomic assessment. This work can substantially fill such gaps. Generally, this manuscript is well-written and well-structured, and fits the scope of this journal. Here are some issues that are suggested to be addressed for further improving this work.

**Response:** We appreciate the Referee's accurate summary and the positive tone. We have one-by-one addressed the Referee's comments.

In Section 3.1, five SSP-RCP combinations were chosen, please briefly discuss the reasons and designs for the combination options, like choosing SSP3-60 and SSP4-70 rather than SSP3-70 and SSP4-60. Since the GCAM-China energy outputs were harmonized by multiplying a balance ratio to eliminate the discrepancy between MEIC and GCAM-China models in the base year. I am concerned about the spatial heterogeneity for the balance ratio during 2010-2015. Could a similar harmonization method be applied for the gap in the base year for CMIP6 and DPEC?

**Response:** We thank the Referee's comments.

First, socio-economic scenarios (i.e. SSPs) constitute an important tool for exploring the long-term consequences of anthropogenic climate change and available response options (Kriegler et al., 2012). As part of the scenario development process, consistent and harmonized quantitative elaborations of population, urbanization and economic development have been developed for all the SSPs. The quantitative elaborations of the SSP narratives are then referred to as 'baseline' scenarios, and the SSP narratives themselves do not include explicit climate policies (Rao et al., 2017). The mitigation effort (i.e. climate mitigation scenarios) of the SSP scenarios is then a function of both the stringency of the target and the underlying energy and carbon intensities in the baselines. This could result in some cases in infeasibilities in terms of meeting mitigation targets for a complete overview of the SSP baseline and climate mitigation scenarios (Rao et al., 2017; Riahi et al., 2017). As shown in Figure R1, not all cells of the matrix have to contain a consistent scenario. For example, a shared socio-economic pathway with rapid development of competitive renewable energies, low population growth and environmental orientation would be hard to reconcile with a 6 degree warming, even without climate policy (Kriegler et al., 2012). The full set of multiple SSPs and forcing outcomes forms a matrix of possible integrated scenarios are shown in Figure R2 (white cells; O'Neill et al., 2016).

Further, in this study, five SSP-RCPs scenarios (i.e. SSP1-26, SSP2-45, SSP3-70, SSP4-60, and SSP5-85 scenarios) are chosen according to the Scenario Model Intercomparison Project (ScenarioMIP) for CMIP6 (Figure R2; O'Neill et al., 2016). O'Neill et al (2016) describe ScenarioMIP's objectives, experimental design, and its relation to other activities within CMIP6 in detail. In summary, O'Neill et al (2016) choose an SSP for each global average forcing pathway by taking into consideration the possibility that the sensitivity of climate outcomes to SSP choice may be larger than anticipated. To

account for that possibility, choices were based on one or, when compatible, more of the following goals: facilitate climate research; minimize differences in climate; and ensure consistency with scenarios that are most relevant to the IAM/IAV communities. Therefore, an experimental design has been identified consisting of eight alternative 21st century scenarios (i.e. SSP5-8.5, SSP3-7.0, SSP2-4.5, SSP1-2.6, SSP4-6.0, SSP4-3.4, SSP5-3.4-OS, SSPa-b) plus one large initial condition ensemble (SSP3-7.0) and a set of long-term extensions (SSP5-8.5-Ext, SSP5-3.4-OS-Ext, and SSP1-2.6-Ext), divided into two tiers defined by relative priority. Therefore, to cover all the SSPs scenarios (one scenario from each SSP), we further choose five abovementioned scenarios from the experimental design in the ScenarioMIP (Gidden et al., 2019). And we have added the related clarification and implications due to such choices in the revised manuscript.

Second, in this work, we harmonized the GCAM-China energy outputs (which have first been reorganized and downscaled to DPEC fuel type categories) by multiplying the base-year balance ratio, which is proceed at the sector and provincial level. That is, GCAM-China collected China's historical data and explored China's future development at the sector and provincial level. Therefore, though the global CMIP6 scenarios are designed at country level, GCAM-China can calibrate and simulate 31 province energy, economic outputs every five years (e.g. 2010 and 2015). The spatial heterogeneity is relatively small. We also checked the base-year (2015, not looking into 2010-2014) energy outputs by province, and revised the evident deviations (i.e. the overestimation of Gansu's biomass use and Beijing's coal use) in GCAM-China. And then we applied the balance ratio of each province to generate future energy, economic outputs. Similar harmonization method could be applied for the gap in the base year for CMIP6 and DPEC, but the spatial heterogeneity is large due to the unavailability of province-level and sector-level data.

[Figure]

**Figure R1.** Matrix of socio-economic "reference" developments (characterized by shared socio-economic pathways, SSPs) and climate change outcomes (determined by representative concentration pathways, RCPs). White cells indicate that not all combinations of shared socio-economic pathways and climate change outcomes may provide a consistent scenario (Kriegler et al., 2012).

[Figure]

**Figure R2.** SSP-RCP scenario matrix illustrating ScenarioMIP simulations (O'Neill et al., 2016).

References:

Kriegler, E., O'Neill, B. C., Hallegatte, S., Kram, T., Lempert, R. J., Moss, R. H. and Wilbanks, T.: The need for and use of socio-economic scenarios for climate change analysis: A new approach based on shared socio-economic pathways, Global Environmental Change, 22(4), 807–822, doi:10.1016/j.gloenvcha.2012.05.005, 2012.

O'Neill, B. C., Tebaldi, C., van Vuuren, D. P., Eyring, V., Friedlingstein, P., Hurtt, G., Knutti, R., Kriegler, E., Lamarque, J.-F., Lowe, J., Meehl, G. A., Moss, R., Riahi, K. and Sanderson, B. M.: The Scenario Model Intercomparison Project (ScenarioMIP) for CMIP6, Geosci. Model Dev., 9(9), 3461–3482, doi:10.5194/gmd-9-3461-2016, 2016.

Riahi, K., van Vuuren, D. P., Kriegler, E., Edmonds, J., O'Neill, B. C., Fujimori, S., Bauer, N., Calvin, K., Dellink, R., Fricko, O., Lutz, W., Popp, A., Cuaresma, J. C., Kc, S., Leimbach, M., Jiang, L., Kram, T., Rao, S., Emmerling, J., Ebi, K., Hasegawa, T., Havlik, P., Humpenöder, F., Da Silva, L. A., Smith, S., Stehfest, E., Bosetti, V., Eom, J., Gernaat, D., Masui, T., Rogelj, J., Strefler, J., Drouet, L., Krey, V., Luderer, G., Harmsen, M., Takahashi, K., Baumstark, L., Doelman, J. C., Kainuma, M., Klimont, Z., Marangoni, G., Lotze-Campen, H., Obersteiner, M., Tabeau, A. and Tavoni, M.: The Shared Socioeconomic Pathways and their energy, land use, and greenhouse gas emissions implications: An overview, Global Environmental Change, 42, 153–168, doi:10.1016/j.gloenvcha.2016.05.009, 2017.

Rao, S., Klimont, Z., Smith, S. J., Van Dingenen, R., Dentener, F., Bouwman, L., Riahi, K., Amann, M., Bodirsky, B. L., van Vuuren, D. P., Aleluia Reis, L., Calvin, K., Drouet, L., Fricko, O., Fujimori, S., Gernaat, D., Havlik, P., Harmsen, M., Hasegawa, T., Heyes, C., Hilaire, J., Luderer, G., Masui, T., Stehfest, E., Strefler, J., van der Sluis, S. and Tavoni, M.: Future air pollution in the Shared Socio-economic Pathways, Global Environmental Change, 42, 346–358, doi:10.1016/j.gloenvcha.2016.05.012, 2017.

Gidden, M., Riahi, K., Smith, S., Fujimori, S., Luderer, G., Kriegler, E., van Vuuren, D.P., van den Berg, M., Feng, L., Klein, D. and Calvin, K. Global emissions pathways under different socioeconomic scenarios for use in CMIP6: a dataset of harmonized emissions trajectories through the end of the century. Geoscientific model development discussions, 12(4), pp.1443-1475, 2019.

Minor comments:

Page2, Line 53: 'is still higher than'

**Response:** We have revised as suggested.

Page2, Line 58: 'benefits substantial reductions', improper use of verb 'benefit'

**Response:** We have replaced "benefits" with "could bring".

Page3, Line 74-75: 'These scenarios describe future emission 75 changes based on a set of scenarios', better to differentiate the 'scenarios' in expression

**Response:** Thanks for the Referee pointing this. We have revised this sentience to "These scenarios describe future emission changes based on a set of assumptions …".

Page3, Line 78: simplify 'do not consider' to 'neglect'?

**Response:** We have revised as suggested.

Page3, Line 80-81: unclear expression 'The absence of public emissions data products makes further use difficult in climate and chemical transport models.'

**Response:** Here we would like to describe that most of the previous scenarios only present the future national emissions in their papers without any public access to further use emissions, not being able to provide gridded emissions as the input of climate and chemical transport models. Therefore, to fill this gap, our study aims to provide a set of emission projection datasets to the community, which has been stated in the manuscript. Here, we have revised this sentence for clarify as suggested.

*"In addition, for other researchers, further studies like air quality and health impact analysis based on these scenarios are difficult to proceed because of the absence of public emissions data products."*

Page3, Line 82-83: confusing expression 'that connects global scenarios considering local policies and representing dynamic emission changes with local policies and technology deployment', rewrite as 'a comprehensive scenario set connecting SSP-RCP scenarios with local policies and representing dynamic emission changes under local policies'?

**Response:** We have revised as suggested.

Page4, Line 96-98: please rephrase this sentence to make it clear

**Response:** We have revised as suggested.

*"The development of this dynamic model and associated scenarios aims to identify win-win measures and pathways to support the future's short and long-term synergizing actions on the environment and climate for the Chinese government."*

Page5, Line 145; Page13, Line 376-377, 394; Page15, Line 449-452: 'Therefore, we used...', better to keep consistency in tense (past tense for methodology)

**Response:** We have revised as suggested.

Page8, Line 219: 'A technology-based emission projection model', remove 's'

**Response:** We have revised as suggested.

Page11, Line 315: 'GCAM-China'

**Response:** We have revised as suggested.

Page12, Line 340 and 349; Page13, Line 391; Page18, Line 542: please pay attention to tense for the clauses of 'we assumed that', 'we found that'

**Response:** Thanks for the Referee pointing this. We have carefully checked the tense of related clauses and revised in the new manuscript.

Page15, Line 435: try to use proper pronoun to make sentence brief

**Response:** We have revised as suggested.

*"Then, four types of fertilizer use are estimated through multiplying the total fertilizer use by their shares in 2015."*

Page18, Line 419: 'is required to be up to'

**Response:** We have revised as suggested.

---

## Author Comment (AC2) · 5 Mar 2020

Due to its substantial energy demand and huge population, China's future air pollutants and greenhouse gas emissions are of great importance both locally and internationally. This work developed a dynamic projection model to predict future emissions under various SSP-RCP scenarios, with a particular focus to integrate local policies. This is an important contribution, and will help to fill the missing gap from those global-scale studies. I suggest minor revisions for acceptance.

**Response:** We appreciate the Referee's accurate summary and the positive tone. We have one-by-one addressed the Referee's comments.

1. SSP and RCP scenarios can interact in different combinations. I understand it is impractical to simulate all scenario combinations, but maybe the author could consider adding some justifications of why choosing SSP1-26, SSP2-45, SSP3-70, SSP4-60, and SSP5-85 scenarios? Also, it would be could to add a few sentences about the implications due to such choices?

[Figure]

**Response:** We thank the Referee for the constructive comments.

First, socio-economic scenarios (i.e. SSPs) constitute an important tool for exploring the long-term consequences of anthropogenic climate change and available response options (Kriegler et al., 2012). As part of the scenario development process, consistent and harmonized quantitative elaborations of population, urbanization and economic development have been developed for all the SSPs. The quantitative elaborations of the SSP narratives are then referred to as 'baseline' scenarios, and the SSP narratives themselves do not include explicit climate policies (Rao et al., 2017). The mitigation effort (i.e. climate mitigation scenarios) of the SSP scenarios is then a function of both the stringency of the target and the underlying energy and carbon intensities in the baselines. This could result in some cases in infeasibilities in terms of meeting mitigation targets for a complete overview of the SSP baseline and climate mitigation scenarios (Rao et al., 2017; Riahi et al., 2017). As shown in Figure R1, not all cells of the matrix have to contain a consistent scenario. For example, a shared socio-economic pathway with rapid development of competitive renewable energies, low population growth and environmental orientation would be hard to reconcile with a 6 degree warming, even without climate policy (Kriegler et al., 2012). The full set of multiple SSPs and forcing outcomes forms a matrix of possible integrated scenarios are shown in Figure R2 (white cells; O'Neill et al., 2016).

Further, in this study, five SSP-RCPs scenarios (i.e. SSP1-26, SSP2-45, SSP3-70, SSP4-60, and SSP5-85 scenarios) are chosen according to the Scenario Model Intercomparison Project (ScenarioMIP) for CMIP6 (Figure R2; O'Neill et al., 2016). O'Neill et al (2016) describe ScenarioMIP's objectives, experimental design, and its relation to other activities within CMIP6 in detail. In summary, O'Neill et al (2016) choose an SSP for each global average forcing pathway by taking into consideration the possibility that the sensitivity of climate outcomes to SSP choice may be larger than anticipated. To account for that possibility, choices were based on one or, when compatible, more of the following goals: facilitate climate research; minimize differences in climate; and ensure consistency with scenarios that are most relevant to the IAM/IAV communities. Therefore, an experimental design has been identified consisting of eight alternative 21st century scenarios (i.e. SSP5-8.5, SSP3-7.0, SSP2-4.5, SSP1-2.6, SSP4-6.0, SSP4-3.4, SSP5-3.4-OS, SSPa-b) plus one large initial condition ensemble (SSP3-7.0) and a set of long-term extensions (SSP5-8.5-Ext, SSP5-3.4-OS-Ext, and SSP1-2.6-Ext), divided into two tiers defined by relative priority.

Therefore, to cover all the SSPs scenarios (one scenario from each SSP), we further choose five abovementioned scenarios from the experimental design in the ScenarioMIP. And we have added the related clarification and implications due to such choices in the revised manuscript.

[Figure]

**Figure R1.** Matrix of socio-economic ''reference'' developments (characterized by shared socio-economic pathways, SSPs) and climate change outcomes (determined by representative concentration pathways, RCPs). White cells indicate that not all combinations of shared socio-economic pathways and climate change outcomes may provide a consistent scenario (Kriegler et al., 2012).

[Figure]

**Figure R2.** SSP-RCP scenario matrix illustrating ScenarioMIP simulations (O'Neill et al., 2016).

References:

Kriegler, E., O'Neill, B. C., Hallegatte, S., Kram, T., Lempert, R. J., Moss, R. H. and Wilbanks, T.: The need for and use of socio-economic scenarios for climate change analysis: A new approach based on shared socio-economic pathways, Global Environmental Change, 22(4), 807–822, doi:10.1016/j.gloenvcha.2012.05.005, 2012.

O'Neill, B. C., Tebaldi, C., van Vuuren, D. P., Eyring, V., Friedlingstein, P., Hurtt, G., Knutti, R., Kriegler, E., Lamarque, J.-F., Lowe, J., Meehl, G. A., Moss, R., Riahi, K. and Sanderson, B. M.: The Scenario Model Intercomparison Project (ScenarioMIP) for CMIP6, Geosci. Model Dev., 9(9), 3461–3482, doi:10.5194/gmd-9-3461-2016, 2016.

Riahi, K., van Vuuren, D. P., Kriegler, E., Edmonds, J., O'Neill, B. C., Fujimori, S., Bauer, N., Calvin, K., Dellink, R., Fricko, O., Lutz, W., Popp, A., Cuaresma, J. C., Kc, S., Leimbach, M., Jiang, L., Kram, T., Rao, S., Emmerling, J., Ebi, K., Hasegawa, T., Havlik, P., Humpenöder, F., Da Silva, L. A., Smith, S., Stehfest, E., Bosetti, V., Eom, J., Gernaat, D., Masui, T., Rogelj, J., Strefler, J., Drouet, L., Krey, V., Luderer, G., Harmsen, M., Takahashi, K., Baumstark, L., Doelman, J. C., Kainuma, M., Klimont, Z., Marangoni, G., Lotze-Campen, H., Obersteiner, M., Tabeau, A. and Tavoni, M.: The Shared Socioeconomic Pathways and their energy, land use, and greenhouse gas emissions implications: An overview, Global Environmental Change, 42, 153–168, doi:10.1016/j.gloenvcha.2016.05.009, 2017.

Rao, S., Klimont, Z., Smith, S. J., Van Dingenen, R., Dentener, F., Bouwman, L., Riahi, K., Amann, M., Bodirsky, B. L., van Vuuren, D. P., Aleluia Reis, L., Calvin, K., Drouet, L., Fricko, O., Fujimori, S., Gernaat, D., Havlik, P., Harmsen, M., Hasegawa, T., Heyes, C., Hilaire, J., Luderer, G., Masui, T., Stehfest, E., Strefler, J., van der Sluis, S. and Tavoni, M.: Future air pollution in the Shared Socio-economic Pathways, Global Environmental Change, 42, 346–358, doi:10.1016/j.gloenvcha.2016.05.012, 2017.

2. The author compared their predicted emissions with CMIP6 results, which reveal notable differences. Could the author provide some relevant guidance/instructions for CMIP6 and future DPEC users? For instance, which model is more reliable/useful under which circumstances? How to interpret the results from these different methods, respectively, in China's context?

**Response:** We thank the Referee for the constructive suggestion.

As we described in the manuscript, our DPEC model aims to provide a set of emission projection datasets, which integrate region-specific and sector-based local policies within the global IPCC scenarios. DPEC is more reliable/suitable for the researchers who focus on the China's near- and mid-term air pollution and climate change issues or look into the global/regional impacts due to China's air pollution and $CO_2$ emissions as their notable differences in recent years. While for other global issues not focusing on China, regional emissions of the CMIP6 results are created under the same frameworks, which can better capture the variations and differences among regions under the same assumptions and circumstances.

In China's context, first, the results from these different methods reveal the differences of historical emission inventory, our MEIC emission inventory can provide more reliable and detailed technology distributions and emissions of the base year, which also lays the foundation of future projections. Second, the differences of future emission trends are because our developed dynamic projection model could better reflect the local policies, which is less considered in the global model.